# Causal effects of dams and land cover changes on flood changes in mainland China

Wencong Yang[1, 2], Hanbo Yang[1,2], Dawen Yang[1,2], and Aizhong Hou[3]

[1] Department of Hydraulic Engineering, Tsinghua University, Beijing 100084, China.
[2] State Key Laboratory of Hydro-Science and Engineering, Tsinghua University, Beijing, 100084, China.
[3] Hydrological Forecast Center, Ministry of Water Resources of the People's Republic of China, Beijing, China

*Correspondence to*: Hanbo Yang (yanghanbo@tsinghua.edu.cn)

**Abstract.** Quantifying the effects of human activities on floods is challenging because of limited knowledge and observations. Many previous methods fail to isolate different effects and reduce the uncertainty caused by small samples. We use panel regressions to derive the sensitivity of annual maximum discharges ($Q$) to the changing values of three human factors: urban areas, cropland areas, and reservoir indexes for large and medium dams. We also test whether the effects increase or decrease with increasing initial values of human factors. This method is applied in 757 non-nested catchments in China. Results show that a 1% point increase of urban areas causes around a 3.9% increase of $Q$ with a confidence interval $CI$=[1.9%, 5.7%]. Cropland areas have no significant effect on $Q$. Reservoir index has a decreasing effect: a 1 unit increase of reservoir index causes a decrease in $Q$ from 21.4% (with $CI$=[11.4%, 29.9%]) to 6.2% (with $CI$=[3.2%%, 9.1%]) for catchments with initial reservoir indexes from 0 to 3. Among 61 catchments with significant increases in observed $Q$ in 1992-2017, increasing urban areas cause more than 10% increases in $Q$ in only 5 (8.2% of 61) catchments. Among 234 catchments with at least one dam and significant decreases in observed $Q$ in 1960-2017, increasing reservoir indexes cause more than 10% decreases in $Q$ in 138 (59.8% of 234) catchments. Among 1249 catchments with limited impacts from urban areas and reservoir indexes, 403 (32.3%) catchments have significant decreases in $Q$ during 1960-2017, and 46.7% of the 403 catchments are located in the middle and down streams of the Yellow River Basin and the upper streams of the Haihe River Basin. This study extends the panel regression method in hydrology and sheds light on the attribution of flood changes on a national scale.

## 1 Introduction

River flooding is one of the most severe disasters in the world. China experiences tremendous damages from floods in the past decades with expanding urban areas, a booming economy, and increasing populations (Du et al., 2019; Kundzewicz et al., 2019). Sharply changing flood characteristics make flood risk management more difficult. According to a national investigation of flood peak changes in China conducted by Yang et al. (2019), abrupt changes due to human activities are the predominant mode of flood changes. Understanding how floods change in a changing environment helps flood risk

management in the future. Therefore, a quantitative attribution of flood changes is urgent on a national scale for policy decisions.

To detect flood changes and pinpoint the underlying reasons, scientists need to answer the following questions: 1. does a factor affect floods? 2. If the effect presents, how strong is the effect? The drivers of flood changes can be classified into three categories: atmospheric factors, catchment factors, and river factors (Merz et al., 2012; Blöschl et al., 2015). Atmospheric factors refer to the meteorological forcing of water fluxes such as natural climate variability and anthropogenic climate change; catchment factors refer to the alternating physiographic characteristics of catchments, such as land cover changes; river factors refer to hydraulic infrastructures that change river morphology and flood routing, such as dams and channelization (Merz et al., 2012; Blöschl et al., 2015). Catchment and river factors are mainly attributed to human activities, which attract increasing attention in hydrological systems in the era of "socio-hydrology" (Di Baldassarre et al., 2019; Müller and Levy, 2019). However, quantifying human impacts on floods is challenging for the following reasons. Firstly, due to the highly unpredictable human behaviors, we have limited knowledge to reproduce the process of how human activities affect floods (Pande and Sivapalan, 2017). For example, the expansion of cropland and urban areas not only casts deterministic effects on floods through changing soil physics and surface roughness but also brings uncertain effects through irrigations and water diversions. We consider these effects "uncertain" because they are related to unknown human decisions. Secondly, the observations of human activities are limited (Pande and Sivapalan, 2017). In the example above, many regions lack long-term and large-scale data of soil physics, roughness, irrigations, and water diversions, which are highly dependent on a high-cost network of in-site measurements.

Previous studies have used three methods to quantify human impacts on floods. The first method is a physical model simulation. This method regards the impacts of human activities as either the difference between actual observations and the model simulations of floods (Viglione et al., 2016; Lu et al., 2018) or flood changes with time-varying model parameters (Peña et al., 2016; Umer et al., 2019). However, this method suffers from limited model accuracy. The second method is a paired-catchment experiment. This method either compares the floods before and after human impacts in one catchment or compares floods in two groups of catchments with and without human impacts (Prosdocimi et al., 2015; Hodgkins et al., 2019). However, the comparisons above cannot rigorously isolate multiple impacts on floods since we cannot actually control everything except one targeted human factor (Runge et al., 2019). The third method is empirical variable dependence, i.e., using regressions or non-stationary probability distributions to link human factors to flood characteristics (FitzHugh et al., 2011; Prosdocimi et al., 2015; Bertola et al., 2019; De Niel and Willems, 2019). The third method is cost-efficient for large-scale studies, but it has two problems. Firstly, to derive the causal effects of human factors, all confounders —— which correlates with human factors and floods at the same time —— should be explicitly accounted for in the empirical relationships. However, defining numerous variables to represent confounders may be an endless task. For example, climatic confounders are ambiguous because floods are caused by different climatic factors (e.g., long rainfall, short rainfall, snowmelt, and rain on snow) in different regions (Stein et al., 2020; Yang et al., 2020a; Merz et al., 2020). Therefore, in a

large-sample study, we do not have a unified regression form to control all possible variables for all catchments. Secondly, empirical methods require sufficient data for robust statistical inference, while flood samples are rare for each catchment.

Panel regression (Steinschneider et al., 2013; Wooldridge, 2016) solves the problems of the empirical method in two ways. Firstly, panel regression adds virtual variables to the regression to represent a fixed individual or regional effect (Steinschneider et al., 2013). In such a way, the regression can account for the effects of ambiguous confounders that are constant in time or region. Secondly, panel regression pools all samples into one model and trades space for time (Steinschneider et al., 2013). Therefore, the regression result is more reliable even with short flood records for each

catchment. Although panel regression is a tool in economics, it has been introduced in hydrology to estimate the effects of forests on floods (Ferreira and Ghimire, 2012), urbanization on runoff coefficients (Steinschneider et al., 2013), dams on streamflow (McManamay et al., 2014), rainfall on streamflow (Bassiouni et al., 2016), deforestation on streamflow (Levy et al., 2018), urbanization on floods (Blum et al., 2020), and rain/snow fraction on floods (Davenport et al., 2020). However, these studies only focused on one factor at one time. Considering more human factors can provide a more comprehensive

picture of the human impacts on floods. In addition, only Blum et al. (2020) and Davenport et al. (2020) tested the nonlinear effects of factors. Previous studies rarely examined whether the effects increase or decrease with increasing initial values of human factors.

In this study, supported by a large dataset of Chinese floods from 757 streamflow gauge stations, we quantify the national average sensitivities of annual flood peaks to changing urban areas, cropland areas, and reservoir indexes for large and

medium dams using panel regression. We also test whether effects increase or decrease for catchments with increasing initial values of the targeted factor. The causal effects of factors distinguish the flood changes explained and unexplained by the three human factors in recent decades. This study is organized as follows. Section 2 introduces methods. Section 3 describes the data. Section 4 presents the results. Section 5 discusses the methods and the insights gained by this study. Section 6 gives conclusions.

## 2 Methods

### 2.1 Causal map of flooding

Causal maps depict the dependency relationship between variables, and they help discover confounders and focus on the causal effects of different factors when fitting a regression model (Pearl and Mackenzie, 2020). A confounder is a variable that influences both human factors and floods. A causal effect is defined as the sensitivity of floods to a factor when all

possible confounding variables are controlled. Similar to Blum et al. (2020), we draw a causal map of flooding in Fig. 1. This study estimates the causal effects of the changes in dams, urban areas, and cropland areas on floods, as the three dashed lines show in the figure. Variables lying above the dashed lines are unknown or unobserved mediators. Urban areas and cropland areas are interrelated because they may change into each other during the process of land cover change. We consider two major confounders. The first confounder is the time-varying confounder, which can be unique for a catchment

or spatially constant in a region. For example, increasing event precipitation during floods, which varies by individual catchments, may promote dam constructions; decreasing annual precipitation, which happens at a regional scale, exacerbates water shortage and may therefore promote the reservoir constructions or the implementation of the Grain for Green Project. We delineate regions by climate since the climate is the first-order driver of catchment similarity (Jehn et al., 2020). In this way, we can control the effect of many omitted variables that have spatial homogeneity. The second confounder is the

individual time-invariant confounder. This confounder is mainly represented by the characteristics of catchment landscapes, e.g., topography, soil types, geology. For example, urban areas are likely built on flat and plain catchments.

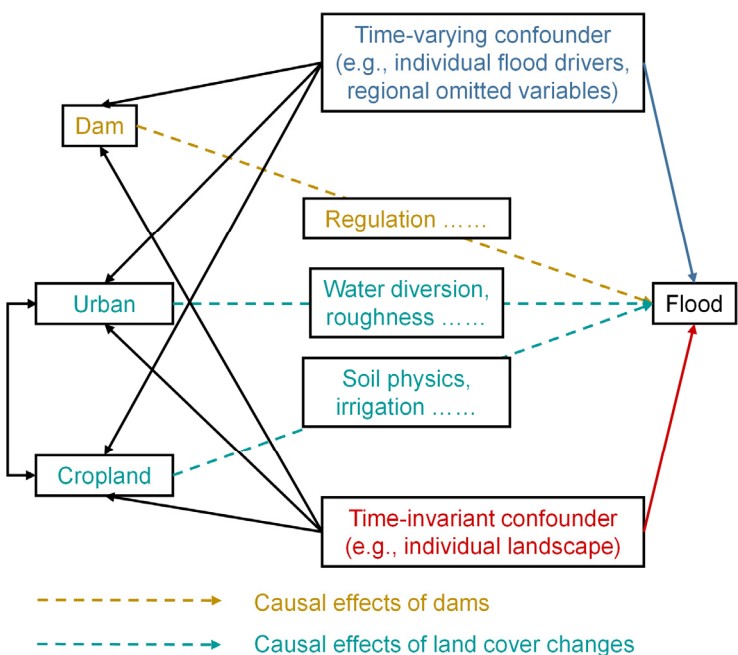

**Figure 1. Causal map illustrating the relationships between human factors and floods.**


## 2.2 Design of panel regression

Panel regression is a statistical technique for panel data (Steinschneider et al., 2013; Wooldridge, 2016). Panel data are observations on several subjects in different periods. Panel regression controls the constant effects of each subject or each period to mitigate regression bias due to omitted variables. Panel regressions in this study are extended from the equation in

Blum et al. (2020) and are presented in Eq. (1) as follows.

$$\log(Q_{i,t}) = \alpha_i + g_1\left(Urban_{i,t}\right) + g_2\left(Crop_{i,t}\right) + g_3\left(RI_{i,t}\right) + \pi_{r,t} D_r D_t + D_r(\varphi_r P_{i,t}^{(3)} + \lambda_r P_{i,t}^{(30)}) + \varepsilon_{i,t} \tag{1}$$

$Q_{i,t}$ is the annual flood peak of catchment $i$ in year $t$ ($m^3 \cdot s^{-1}$). $Urban_{i,t}$ is the urban percentage of catchment $i$ in year $t$ (%). $Crop_{i,t}$ is the cropland percentage of catchment $i$ in year $t$ (%). $RI_{i,t} = \sum_j \left( A_i^{(j)}/A_i \right) \cdot DOR_i^{(j)}$, is the reservoir index of catchment $i$ in year $t$; $DOR_i^{(j)}$ is the degree of regulation of reservoir $j$ in catchment $i$, which is the ratio between the storage capacity and total annual flow of the reservoir; $A_i^{(j)}$ is the upstream area of reservoir $j$; $A_i$ is the area of catchment $i$. $D_r$ is a region dummy which equals 1 or 0. $D_t$ is a year dummy which equals 1 or 0. $P_{i,t}^{(3)}$ is the 3-day total precipitation before the flood peak in year $t$ of catchment $i$, which accounts for the rainfall that causes the flood; $P_{i,t}^{(30)}$ is the 30-day total precipitation before the flood peak in year $t$ of catchment $i$, which accounts for the soil moisture and snowmelt that cause the flood. The coefficients of $P_{i,t}^{(3)}$ and $P_{i,t}^{(30)}$, namely $\varphi_r$ and $\lambda_r$, are assumed to be constant within a climatic region $r$. $\alpha_i$ is the time-invariant constant effects of catchment $i$. $\pi_{r,t}$ is the constant effects of region $r$ in year $t$. $\varepsilon_{i,t}$ is the model residuals. The response functions $g_1(\cdot)$, $g_2(\cdot)$, and $g_3(\cdot)$ represent various response types of $Q$ to different factors.

A region consists of a group of spatially coherent catchments with a similar climate. Unlike Blum et al. (2020) who used predefined physiographic regions, we delineated regions by using the partitioning around medoids (PAM) algorithm (Reynolds et al., 2006) based on the distance matrix of all catchments defined as follow:

$$dist(i,j) = dist_{KG}(i,j) + dist_{cen}(i,j) \tag{2}$$

$$dist_{KG}(i,j) = \frac{1}{2}\sum_{l=1}^{30} |k_i^{(l)} - k_j^{(l)}| \tag{3}$$

$$dist_{cen}(i,j) = \left( \frac{earth.dist(i,j)}{\max\{earth.dist(i,j)|\forall i,j\}} \right)^{1/2} \tag{4}$$

where $dist_{KG}(i,j)$ is the distance of Köppen–Geiger class (Beck et al., 2018) ratios between catchment $i$ and $j$; $k_i^{(l)}$ is the area percentage of Köppen–Geiger class $l$ in catchment $i$; $dist_{cen}(i,j)$ is the standardized distance between the geometric centers of catchment $i$ and $j$; $earth.dist(i,j)$ is the spherical distance on the earth between the geometric centers of catchment $i$ and $j$.

The effect of a factor $X$ on $Q$, i.e., the percentage change in $Q$ given a fixed change in $X$, is expressed as:

$$\Delta Q(\%) = \Delta Q/Q = \exp(g(X + \Delta X) - g(X)) - 1 \tag{5}$$

We considered three types of response functions $g(\cdot)$. $g(X_{i,t}) = \beta X_{i,t}$ indicated a stable effect where the percentage change in $Q$ only depended on $\Delta X$; $g(X_{i,t}) = \gamma X_{i,t}^2$ indicated an increasing effect where the percentage change in $Q$ increased with increasing $X_{i,t}$; $g(X_{i,t}) = \theta X_{i,t}^{1/2}$ indicated a decreasing effect where the percentage change in $Q$ decreased with increasing $X_{i,t}$. To determine the specific effect type, we fitted regressions with 27 possible combinations of $[g_1(\cdot), g_2(\cdot), g_3(\cdot)]$ types and selected the one with the lowest AIC value. We used bootstrapping to test the significance and derive the confidence

intervals of coefficients $\beta$, $\gamma$, and $\theta$ so that the model residuals were allowed to be non-Gaussian and the sampling uncertainty could be accounted for.

The mathematical assumptions of the panel regressions in this study are as follows: 1. There are no other time-varying sub-regional variables that correlate with both human factors and floods; 2. There are no interactions between human factors and regional or individual characteristics that produce significant spatially heterogeneous effects. The regressions and statistical tests were performed in R (R Core Team, 2019) using packages lfe (Gaure, 2019).

## 2.3 Flood change quantification

To examine whether the changes in observed floods can be explained by the changes in human factors, we first detected catchments with significant changes in $Q$ using Mann-Kendall test (Mann, 1945) and Pettitt's test (Pettitt, 1979), and then derived the accumulated flood changes attributed to the change in factor $X$ for catchment $i$ from year $t_1$ to $t_2$:

$$\Delta Q(\%) = \exp(g(X_{i,t_2}) - g(X_{i,t_1})) - 1 \tag{6}$$

To examine how floods changed in catchments that were free from the impacts of urban areas, cropland areas, and dams, we selected catchments with less than 10% changes in flood peaks due to those factors respectively. Specifically, for a factor $X$, we selected catchments with $\left|\exp\left(g(X_{i,t_2}) - 0\right) - 1\right| < 10\%$ where $t_2$ was the most recent year of the data. Then, we applied the Mann-Kendall test (Mann, 1945) and Pettitt's test (Pettitt, 1979) in those catchments to detect the ones with significant changes in $Q$.

## 3 Data

### 3.1 Streamflow and precipitation data

Annual maximum instantaneous discharge data in 2739 streamflow gauge stations were obtained from the Ministry of Water Resources in China (http://www.mwr.gov.cn/english/). Figure 2 shows the outlet locations of all stations. The catchment areas are from 1 km² to 1,705,383 km² with a median of 1,660km². Catchment boundaries were extracted using MERIT-Hydro hydrography data (Yamazaki et al., 2019). Differences between extracted catchment areas and reported areas were less than 20% for all catchments. We only used data from 1960 to 2019 because less than 1,000 stations had available data before 1960. Notice that a few stations in the northeast lie outside mainland China. They were not excluded from this study because all other data were globally available. The 1-km resolution data of Köppen-Geiger climate classes were obtained from Beck et al. (2018). A 3-hourly and 0.1° precipitation dataset in 1979-2017, the Multi-Source Weighted-Ensemble Precipitation Version 2.2 (MSWEP V2.2; Beck et al., 2019), was used.

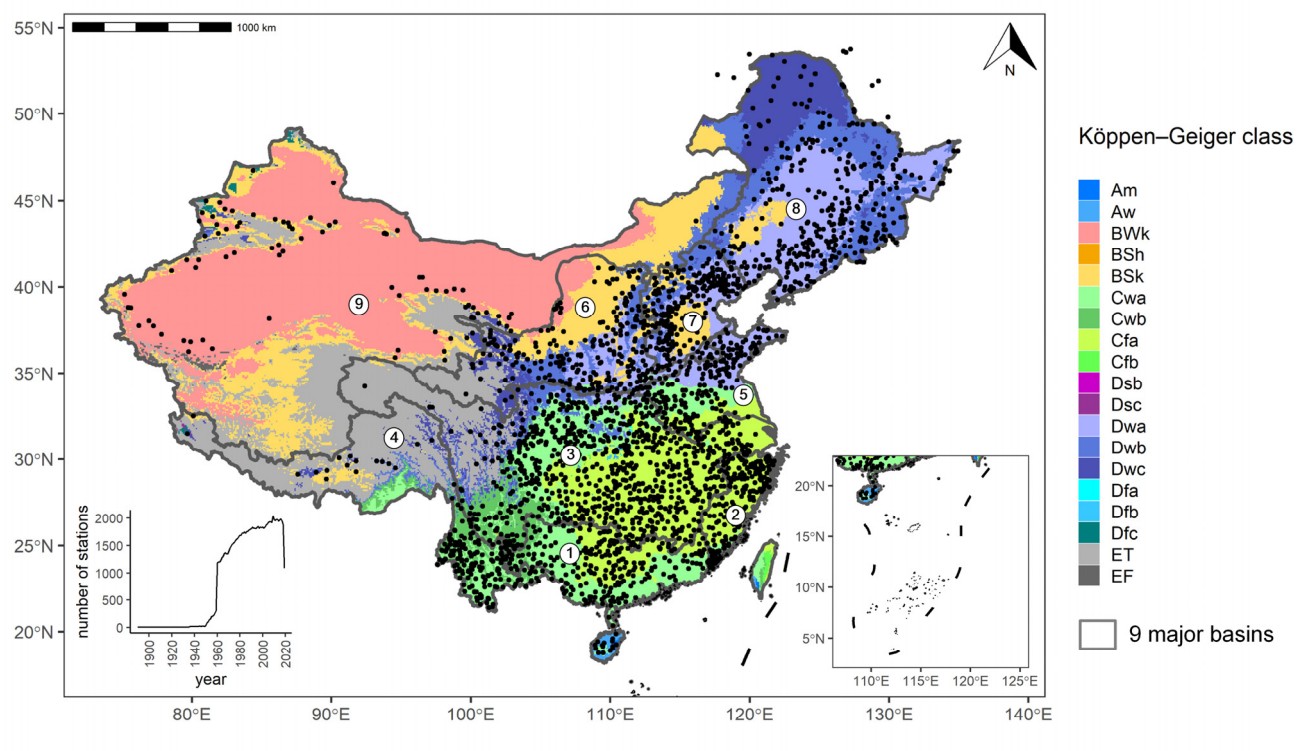

**Figure 2. National 2739 streamflow stations and the number of stations with available annual maximum discharges each year. The Köppen-Geiger climate classes are obtained from Beck et al. (2018). The boundary lines delineate nine major river basins of China: 1. the Pearl River Basin, 2. the Southeast Basin, 3. the Yangtze River Basin, 4. the Southwest Basin, 5. the Huaihe River Basin, 6. the Yellow River Basin, 7. the Haihe River Basin, 8. the Songliao River Basin, and 9. the Continental Basin.**

## 3.2 Land cover and dams data

Land cover maps were obtained from the CCI-LC product produced by the European Space Agency (ESA) Climate Change Initiative (CCI). This product provides global yearly 300m-resolution land cover data in 1992-2015 in version 2.0.7 and 2016-2018 in version 2.1.1 (http://maps.elie.ucl.ac.be/CCI/viewer/download.php). Urban areas of catchments can be extracted from the maps directly. Cropland areas consist of rain-fed cropland, irrigated or post-flooding cropland, and mosaic cropland.

Dam data were available in the Global Reservoir and Dam (GRanD) v1.3 database (Lehner et al., 2011). GRanD collected information about 7,320 global dams in 1948-2017 and recorded 923 dams with storage capacities larger than 10 million $m^3$ in China. These 923 dams were categorized as large and medium dams, according to the Bulletin of the first National Water Conservancy Survey (http://www.chinawater.com.cn/ztgz/xwzt/2013slpczt/1/). The total storage capacity of all 4,694 large and medium dams is 861,961 million $m^3$ in China, according to the bulletin. The total storage capacity of all the 923 GRanD

dams in China is 670,158 million m$^3$, approximately 78% of that recorded in the bulletin. It suggests that the GRanD
database is reliable for quantifying the effects of large and medium dams on floods while it is unsuitable for considering
small dams. For simplicity, we use "dams" to represent large and medium dams in the rest of the paper. We obtained the
locations, upstream areas, storage capacities, and total annual flows of each dam from the database. The reservoir index can
be calculated with the information above.

### 3.3 Catchment selection for regression setup

We selected catchments with at least 20 years of annual maximum discharges ($Q$) to fit Eq. (1) in the common period of
CCI-LC and GRanD data, i.e., 1992-2017. To avoid inaccurate statistical inference on the regression coefficients due to the
correlated model residuals caused by nested catchments, we selected the most upstream catchments with large or medium
dams (if possible) among overlapping catchments. We got 757 independent (non-nested) catchments, among which 207
catchments had as least one dam. The statistics of catchment characteristics in those 757 catchments are presented in Table 1
and the spatial distribution of catchments is presented in Fig. 3. Catchments with changes in $Urban$, $Crop$, and $RI$ have
large impacts on estimating regression coefficients. The numbers of catchments with $\Delta Urban > 0$, $\Delta Crop > 0$, and $\Delta RI > 0$
are 656, 351, and 64, respectively. The number of catchment groups $k$ in Section 2.2 had no optimal value. In light of the
number of selected catchments in the regression models, we set $k$ to be 10, 20, 30, 40, 50, 60, 70, and 80 to test the
robustness of the models.

**Table 1. Summary of catchment characteristics for 757 independent catchments. For each catchment, among its all years with
available flood data in 1992-2017, we choose the last year to calculate *Urban*, *Crop*, and *RI*, and choose the first and last year to
calculate ∆*Urban*, ∆*Crop*, and ∆*RI*. The summaries of *RI* and ∆*RI* are calculated based on 207 catchments with at least one large
and medium dam.**

| Variables | Min. | 1st Qu. | Median | Mean | 3rd Qu. | Max. |
|---|---|---|---|---|---|---|
| *Area* ($km^2$) | 29 | 499 | 1096 | 3341 | 2763 | 142372 |
| *Urban* (%) | 0 | 0.06 | 0.30 | 1.52 | 1.10 | 65.07 |
| ∆*Urban* (%) | 0 | 0.05 | 0.23 | 1.14 | 0.85 | 24.66 |
| *Crop* (%) | 0 | 10.63 | 24.71 | 32.75 | 48.99 | 99.58 |
| ∆*Crop* (%) | -21.58 | -0.81 | -0.02 | 0.38 | 0.87 | 32.04 |
| *RI* | 0.01 | 0.09 | 0.21 | 0.51 | 0.61 | 7.45 |
| ∆*RI* | 0 | 0 | 0 | 0.17 | 0.07 | 7.44 |

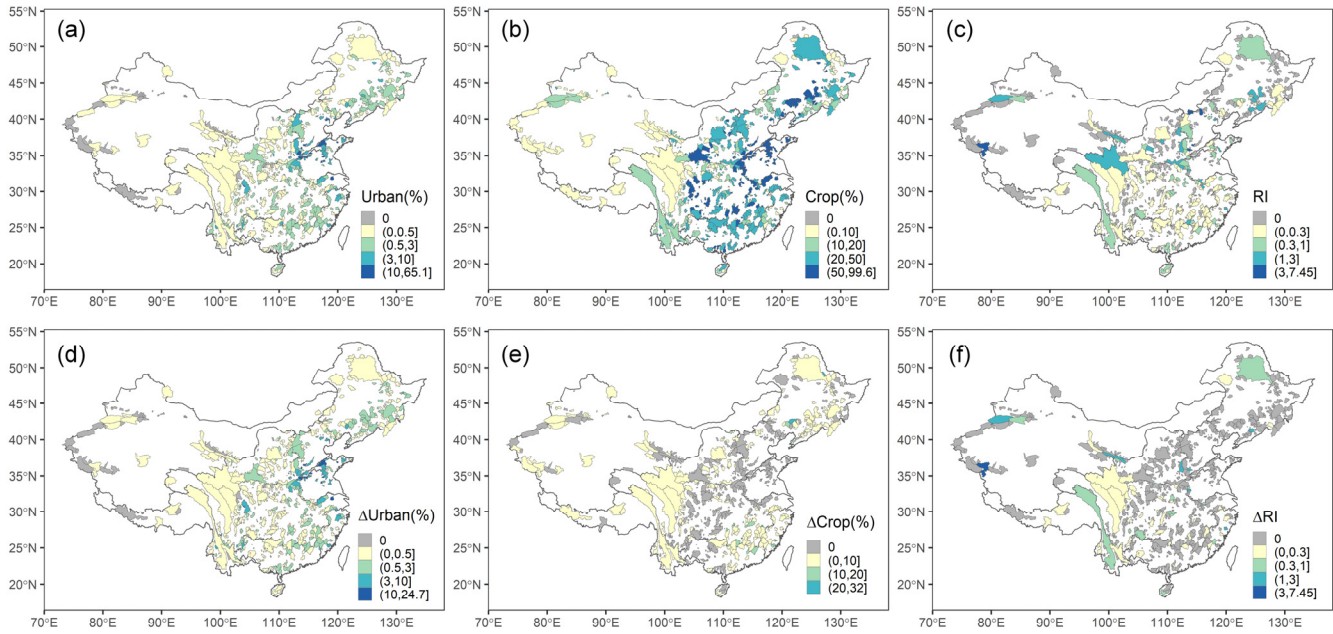

**Figure 3. Spatial distribution of catchment characteristics in 757 independent catchments. (a) Urban percentages ($Urban$), (b) cropland percentages ($Crop$), and (c) reservoir indexes ($RI$) in their last years with available flood data. The changes of (d) urban areas ($\Delta Urban$), (e) cropland areas ($\Delta Crop$), and (f) reservoir indexes ($\Delta RI$) in 1992-2017.**

## 4 Results

### 4.1 The sensitivity of floods to human factors

Table 2 shows the optimal regression forms according to AIC and the corresponding coefficient estimations of factor $Urban$, $Crop$ and $RI$ in Eq. (1). For $Urban$ and $RI$, the effect types are consistent for varying values of $k$. $Urban$ exhibits a positive
and stable effect ($g(X_{i,t}) = \beta X_{i,t}$ and $\beta > 0$), which means a fixed percentage point increase of $Urban$ brings a fixed percentage increase of $Q$ no matter how large the initial $Urban$ is. $RI$ exhibits a negative and decreasing effect ($g(X_{i,t}) = \theta X_{i,t}^{1/2}$ and $\theta < 0$), which means a fixed increase of $RI$ brings a lower percentage decrease of $Q$ with a larger value of initial $RI$. $Crop$ exhibits no significant effect. The maps of catchment groups for all $k$ values can be seen in Fig. A1 of Appendix A. Figure 4 shows the percentage change in $Q$ caused by a 1% point increase of urban area according to Eq. (5). The values of
$\Delta Q(\%)$ are relatively consistent with varying values of $k$. $Crop$ has no significant effect on $Q$, therefore we do not calculate the corresponding sensitivity. Figure 5 shows the percentage change in $Q$ caused by a 1 unit increase of $RI$ according to Eq. (5). The relationship between $\Delta Q$ and $RI$ has little change when $k \geq 20$. In summary, the method is robust to $k$, and thus, we regard the model with $k = 50$ as the main model in the remaining part of the study. The sensitivity of $Q$ to $Urban$ is $\Delta Q =$

3.9% with the 95% confidence interval $CI = [1.9\%, 5.7\%]$ when $\Delta Urban = 1\%$. The absolute values of $\Delta Q$ decrease with increasing initial values of $RI$. For initial $RI = 0$, $\Delta Q = -21.2\%$ with $CI = [-29.9\%, -11.4\%]$ when $RI$ increases by 1; for initial $RI = 3$, $\Delta Q = -6.2\%$ with $CI = [-9.1\%, -3.2\%]$ when $RI$ increases by 1.

**Table 2. Optimal regression forms (with the lowest AIC) and the corresponding coefficient estimations of factor urban percentage ($Urban$), cropland percentage ($Crop$), and reservoir index ($RI$) in Eq. (1). * denotes that the coefficient does not equal to 0 at a 0.05 significance level using bootstrapping inference. $k$ is the preset number of catchment groups.**

| | $k$ | Optimal $g(\cdot)$ | 2.5% Qu. | Mean | 97.5% Qu. |
|---|---|---|---|---|---|
| | 10 | $\beta Urban_{i,t}$ * | 1.20E-02 | 2.72E-02 | 4.21E-02 |
| | 20 | $\beta Urban_{i,t}$ * | 2.31E-02 | 3.89E-02 | 5.59E-02 |
| | 30 | $\beta Urban_{i,t}$ * | 9.35E-03 | 2.65E-02 | 4.40E-02 |
| | 40 | $\beta Urban_{i,t}$ * | 1.59E-02 | 3.64E-02 | 5.68E-02 |
| *Urban* effect | 50 | $\beta Urban_{i,t}$ * | 1.88E-02 | 3.71E-02 | 5.57E-02 |
| | 60 | $\beta Urban_{i,t}$ * | 1.08E-02 | 3.10E-02 | 5.03E-02 |
| | 70 | $\beta Urban_{i,t}$ * | 1.32E-02 | 3.18E-02 | 5.05E-02 |
| | 80 | $\beta Urban_{i,t}$ * | 1.22E-02 | 3.23E-02 | 5.27E-02 |
| | 10 | $\theta Crop_{i,t}^{1/2}$ | -6.14E-02 | 1.43E-02 | 8.87E-02 |
| | 20 | $\theta Crop_{i,t}^{1/2}$ | -4.59E-02 | 2.95E-02 | 1.01E-01 |
| | 30 | $\gamma Crop_{i,t}^{2}$ | -1.41E-04 | -5.36E-05 | 3.42E-05 |
| | 40 | $\gamma Crop_{i,t}^{2}$ | -1.36E-04 | -3.81E-05 | 5.61E-05 |
| *Crop* effect | 50 | $\theta Crop_{i,t}^{1/2}$ | -5.21E-02 | 2.49E-02 | 1.09E-01 |
| | 60 | $\gamma Crop_{i,t}^{2}$ | -1.66E-04 | -7.54E-05 | 2.77E-05 |
| | 70 | $\gamma Crop_{i,t}^{2}$ | -1.62E-04 | -6.71E-05 | 2.85E-05 |
| | 80 | $\gamma Crop_{i,t}^{2}$ | -1.71E-04 | -7.21E-05 | 2.74E-05 |
| | 10 | $\theta RI_{i,t}^{1/2}$ * | -2.90E-01 | -1.74E-01 | -5.84E-02 |
| | 20 | $\theta RI_{i,t}^{1/2}$ * | -3.43E-01 | -2.11E-01 | -1.01E-01 |
| | 30 | $\theta RI_{i,t}^{1/2}$ * | -3.48E-01 | -2.20E-01 | -1.08E-01 |
| | 40 | $\theta RI_{i,t}^{1/2}$ * | -3.64E-01 | -2.45E-01 | -1.25E-01 |
| *RI* effect | 50 | $\theta RI_{i,t}^{1/2}$ * | -3.55E-01 | -2.39E-01 | -1.21E-01 |
| | 60 | $\theta RI_{i,t}^{1/2}$ * | -3.85E-01 | -2.55E-01 | -1.30E-01 |
| | 70 | $\theta RI_{i,t}^{1/2}$ * | -3.80E-01 | -2.38E-01 | -1.14E-01 |
| | 80 | $\theta RI_{i,t}^{1/2}$ * | -3.88E-01 | -2.36E-01 | -9.03E-02 |

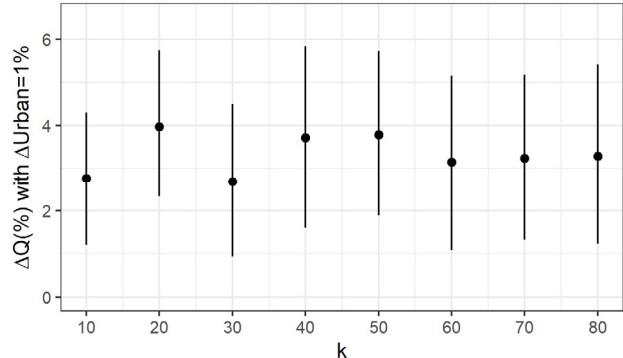

**Figure 4. Percentage change in annual maximum discharge ($Q$) caused by a 1% point increase of urban area ($Urban$) based on different numbers of catchment groups ($k$). The error bars are 95% confidence intervals.**

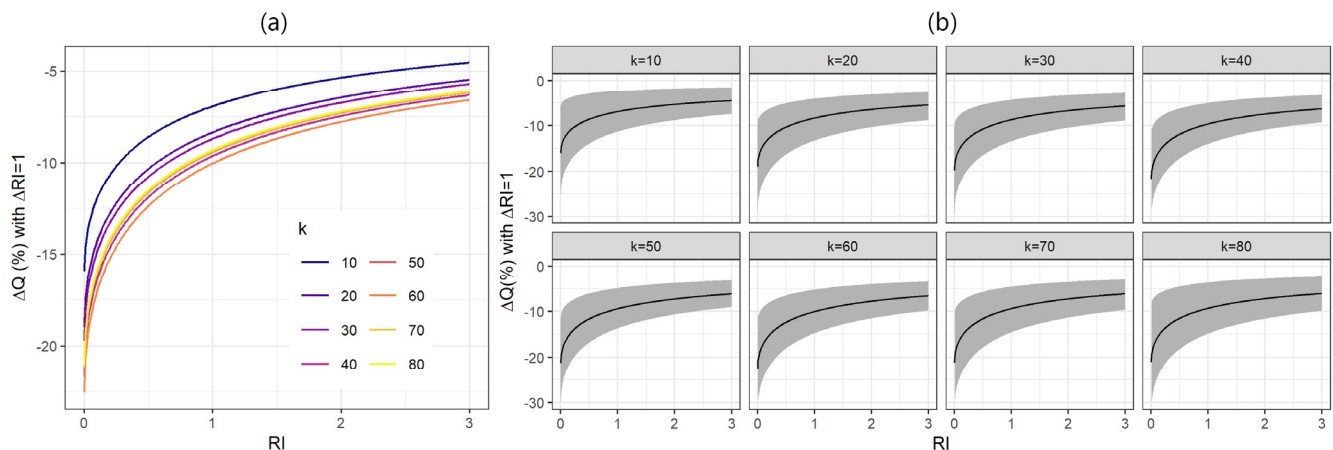

**Figure 5. (a) Percentage change in annual maximum discharge ($Q$) caused by a 1 unit increase of reservoir index ($RI$) from different initial $RI$ values based on different numbers of catchment groups ($k$). (b) The same as (a) but with the 95% confidence intervals shown by shaded areas.**

## 4.2 National flood changes and attributions

Figure 6 shows the changes in $Q$ due to the changes in $Urban$ for 1625 catchments with at least 20 years of data in 1992-2017, according to Eq. (6). To avoid sample heterogeneity between these 1625 catchments and the 757 catchments used for regression, we compared the frequency distribution of $Urban$ and $\Delta Urban$ for catchments with $\Delta Urban > 0$ in the two samples of catchments in Fig. A2. Since there is no substantial difference between the two distributions in Fig. A2, the sensitivity of $Q$ to $Urban$, which is derived from the 757 catchments, can be used to infer the changes in $Q$ due to $\Delta Urban$

in those 1625 catchments. According to Fig. 6, increasing *Urban* causes increases in *Q* of more than 10% in 184 (11.3% of 1625) catchments, which are mainly located in the North Plain of China, especially in the Huaihe River Basin and the middle and down streams of the Haihe River Basin. Among these 184 catchments, increasing *Urban* causes increases in *Q* of more than 25% in 71 (4.4% of 1625) catchments. In 61 catchments with significant increases in observed *Q*, increasing *Urban* causes increases in *Q* of more than 10% in only 5 catchments, which means urbanization is not a predominant driver of flood changes in 1992-2017 in China. The changes in *Q* due to the changes in *Crop* cannot be effectively quantified because *Crop* has no statistically significant effects on *Q* according to the results in Section 4.1.

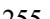

**Figure 6. Accumulated increases in annual maximum discharges (*Q*) due to the increases in urban areas (*Urban*) for 1625 catchments with at least 20 years of flood data in 1992-2017, according to Eq. (6). Large dots represent catchments with significant increases in observed *Q* if p<0.05 for any one of the Mann-Kendall test and the Pettitt's test. The boundary lines delineate nine major river basins of China.**

Figure 7 shows the changes in $Q$ due to the changes in $RI$ for 536 catchments with at least one dam and at least 30 years of data in 1960-2017, according to Eq. (6). Similar to Fig. 6, to avoid sample heterogeneity between these 536 catchments and the 757 catchments used for regression, we compared the frequency distribution of $RI$ and $\Delta RI$ for catchments with $\Delta RI > 0$

in the two samples of catchments in Fig. A3. Since there is no substantial difference between the two distributions in Fig. A3, the sensitivity of $Q$ to $RI$, which is derived from the 757 catchments, can be used to infer the changes in $Q$ due to $\Delta RI$ in those 536 catchments. According to Fig. 7, in 196 (36.6% of 536) catchments, increasing $RI$ leads to more than 10% decreases in $Q$. It indicates that flood peaks are likely to decrease severely if dams are built in the catchment. Among these 196 catchments, increasing $RI$ leads to more than 25% decreases in $Q$ in 28 (5.2% of 536) catchments. Spatially, the impacts

of dams on floods are larger in northern basins (the Huaihe River Basin, the Haihe River Basin, the Yellow River Basin, and the Songhua and Liaohe River Basin) than that in southern basins (the Yangtze River Basin, the Southeast River Basin, the Southwest River Basin, and the Pearl River Basin). In the northern basins, increasing $RI$ leads to more than 10% and 25% decreases in $Q$ in 47.8% and 10.0% catchments, respectively. By comparison, in southern basins, increasing $RI$ leads to more than 10% and 25% decreases in $Q$ in only 26.3% and 0.7% catchments, respectively. In 234 catchments with significant

decreases in observed $Q$, increasing $RI$ leads to more than 10% decreases in $Q$ in 138 (59.0% of 234) catchments, which means dam construction is a predominant driver of the decreases in flood magnitudes for catchments with dams in 1960-2017.

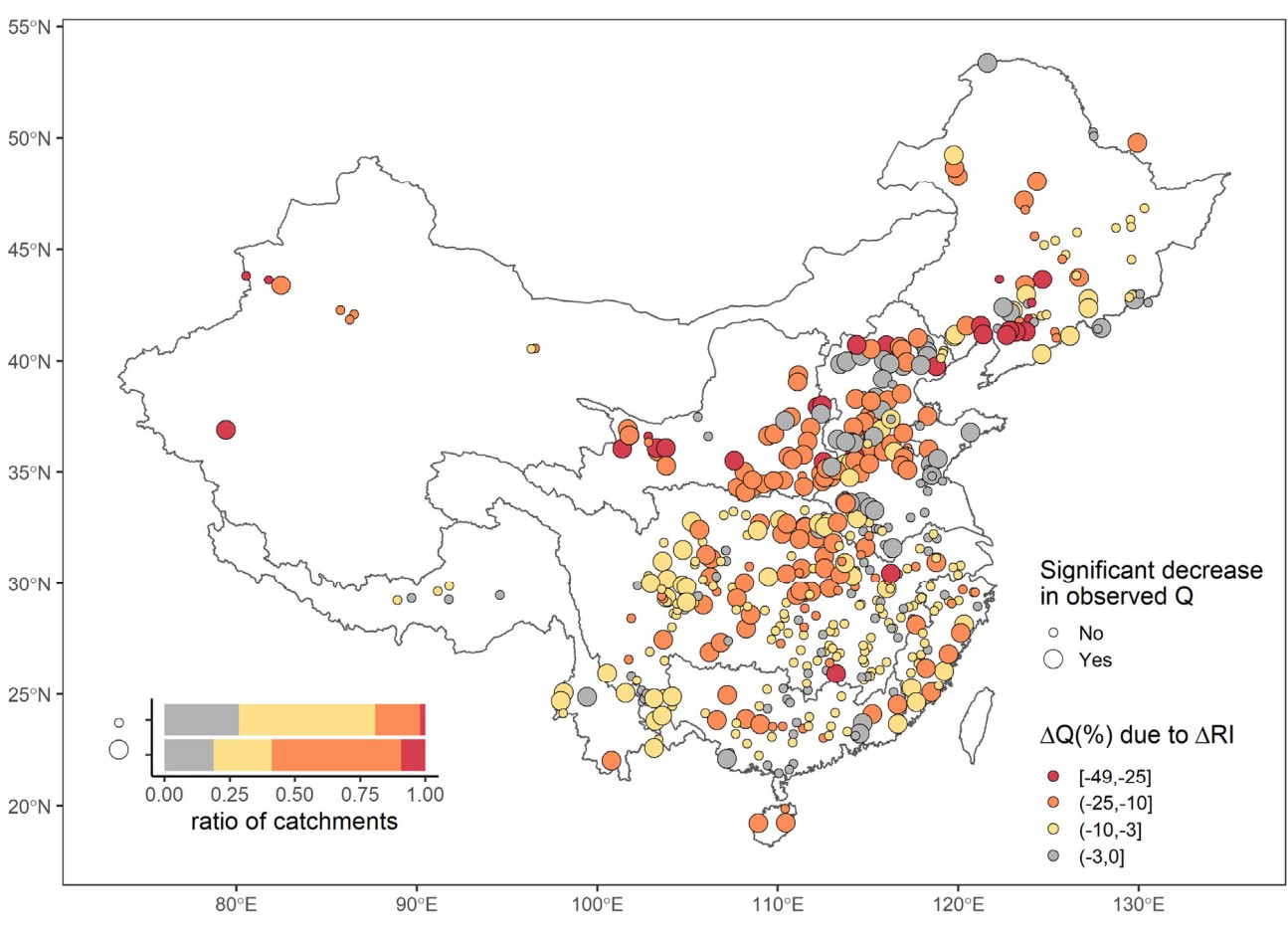

**Figure 7. Accumulated decreases in annual maximum discharges ($Q$) due to the increases in reservoir index ($RI$) for 536 catchments with at least 30 years of flood data and at least one dam in 1960-2017, according to Eq. (6). Large dots represent catchments with significant decreases in observed $Q$ if p<0.05 for any one of the Mann-Kendall test and the Pettitt's test. The boundary lines delineate nine major river basins of China.**

Figure 8 shows the change directions of $Q$ during 1960-2017 in 1249 catchments with at least 30 years of data, $Urban <$ 2.6% and $RI < 0.19$. These catchments were selected based on the thresholds that ensure each factor leads to no more than 10% changes in $Q$, as stated in Section 2.3. In these catchments, the changes in $Q$ are free from the impacts of $Urban$, $Crop$, and $RI$. Significant increases in $Q$ occur in 85 (6.9% of 1249) catchments. Significant decreases in $Q$ occur in 403 (32.3% of 1249) catchments, among which 188 (46.7% of 403) are located in the Yellow River Basin (mainly in the middle and down streams) and the Haihe River Basin (mainly in the upper streams). Such regional coherence of similar trends cannot be found in other regions.

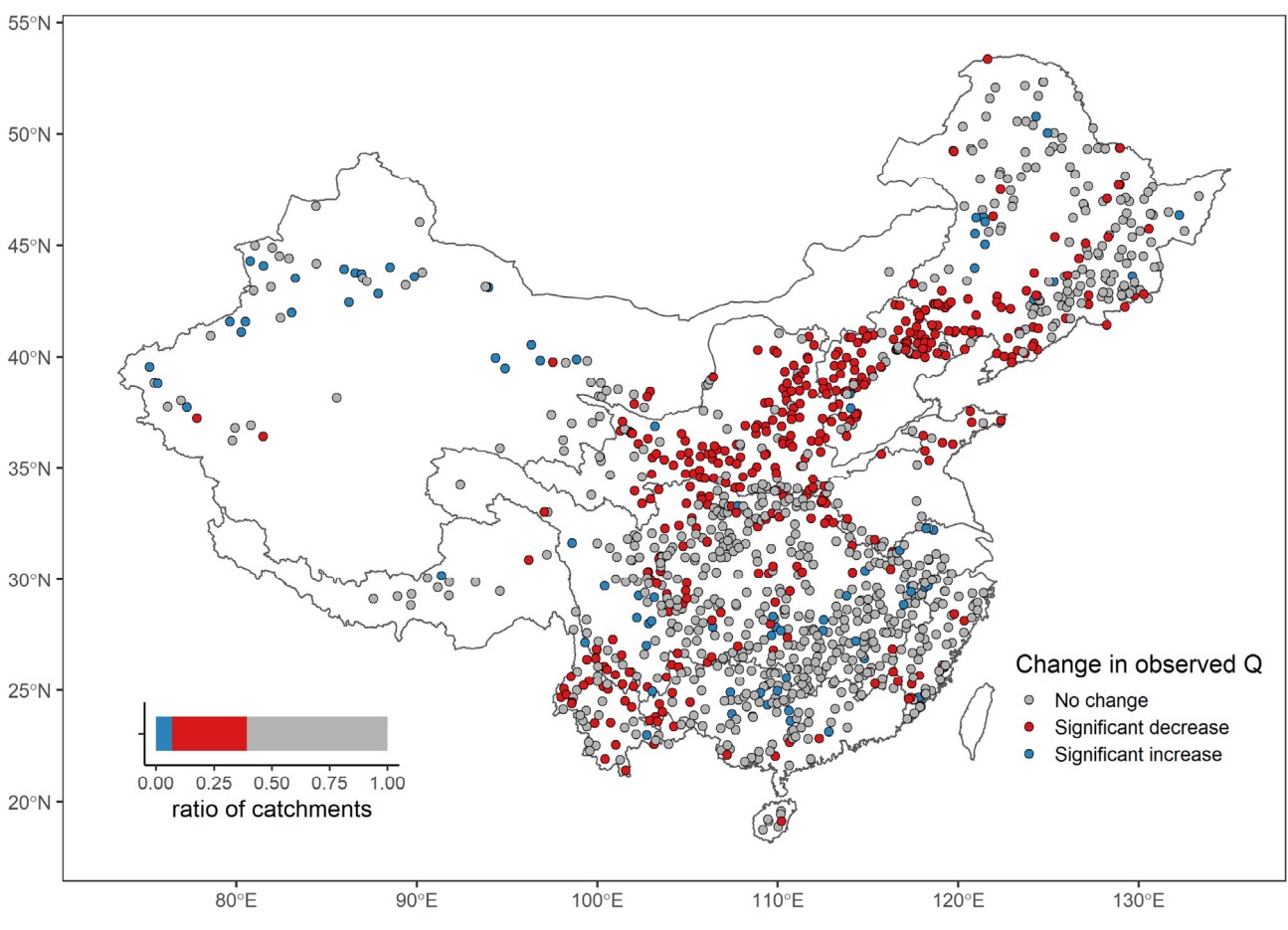

Figure 8. Change directions of annual maximum discharges ($Q$) during 1960-2017 for 1249 catchments with at least 30 years of flood data, $Urban < 2.6\%$ and $RI < 0.19$. These catchments are considered to be free from the impacts of urbanization and dam constructions. The change is significant if p<0.05 for any one of the Mann-Kendall test and Pettitt's test. The boundary lines delineate nine major river basins of China.

## 5 Discussion

### 5.1 Strengths and limitations of panel regressions

We use panel regressions to derive the causal effects of urban areas, cropland areas, and dams on annual maximum discharges across mainland China. In this study, the panel regressions exhibit the following strengths. 1. We obtain a

nationally generalizable sensitivity of floods to each human factor. This sensitivity helps understand the overall added risks of specific human activity on floods on a national scale. In addition, with quantitative sensitivity, scientists are able to select catchments with limited impacts of dams and land cover changes for studying the effects of climate change, e.g., Blöschl et al. (2019). 2. Compared with previous studies using panel regressions in hydrology (Ferreira and Ghimire, 2012; Steinschneider et al., 2013; McManamay et al., 2014; Bassiouni et al., 2016; Levy et al., 2018; Blum et al., 2020; Davenport

et al., 2020), we take a further step by considering multiple types of human impacts simultaneously and distinguishing their increasing or decreasing effects. Blum et al. (2020) and Davenport et al. (2020) considered non-linear forms of response functions for the targeted factors, but they did not distinguish increasing and decreasing effects. These improvements provide a more comprehensive understanding of human impacts on floods.

   The limitations are as follows. 1. The assumptions in the regressions are difficult to test. As stated in Section 2, we assume (i)

no more important time-varying sub-regional confounders and (ii) no interaction terms between human factors and regional or individual characteristics that produce significant spatially heterogeneous effects. These assumptions may be violated in some cases. For example, the effect of urbanization on floods may be larger in regions with higher soil permeability, which means spatially heterogeneous effects may be nonnegligible. Testing these assumptions requires detailed information about catchment characteristics such as topography and geology. Moreover, adding too many variables into the regressions will

decrease model interpretability. 2. The method cannot distinguish the heterogeneous effects of human factors on different floods. As stated in Section 2.3, the method derives a common percentage change in all flood peaks given changing human factors, which means no changes in coefficients of variation. However, practically, the variability of floods may change by human activities. For example, reservoirs tend to regulate extreme floods but omit small floods. 3. This study does not comprehensively assess the effects of total human impacts on floods. We omit many other human factors due to the lack of

data. For example, the data about water diversion, irrigation, channelization, and afforestation on a national scale are currently not available to the public.

## 5.2 Consistency with knowledge and other large-sample studies

   We detect a stable positive effect of urban areas on floods. In theory, expanding urban areas magnify floods in two major ways. Firstly, natural soil grounds are replaced by impervious surfaces, which lead to more rainfall water appearing on the

surface rather than infiltrating into the soil (Villarini and Slater, 2017). Second, urban areas have smooth surfaces, where floods propagate faster and become more flashy (Mogollón et al., 2016). This study finds a 3.9% (with $CI = [1.9\%, 5.7\%]$) increase in annual maximum discharges given a 1% point increase in urban areas. This finding accords with the result from a national investigation in the US (Blum et al., 2020), which reported a 3.3% (with $CI = [1.9\%, 4.7\%]$) increase in annual maximum discharges based on panel regressions.

The cropland areas impose no significant impacts on floods according to our results. Theoretically, expanding cropland areas affect floods in many ways. For example, during agricultural practices, soil depths may decrease due to erosion while increase due to soil compaction (Rogger et al., 2017). Cropland may also bring artificial drainages that lower groundwater

tables (Rogger et al., 2017). Some effects may be offset by others, which masks the relationship between cropland areas and floods. Similar to our result, Bertola et al. (2019) found that agricultural land-use intensification rarely caused flood changes in 95 catchments of Austria using covariate-based non-stationary flood probability distributions. To our knowledge, large-sample studies are limited on the relationship between cropland and floods. Therefore, more detailed in-site investigations are required to uncover the causal chain from cropland changes to flood changes.

This study suggests that dams have a negative decreasing effect on floods. Generally, dams buffer water during floods and thus decrease flood peaks. More dams may not necessarily decrease floods at a constant rate because existing dams with sufficient storage capacities are already capable to control floods. This effect was confirmed by Wang et al. (2017), who used detailed conceptual models of reservoir regulation and found that the mean annual floods had a slowing decrease with increasing reservoirs. In a large-sample study on 4859 catchments in the US (FitzHugh et al., 2011), median annual 1-day maximum flows were estimated to decrease by more than 20% when the storage ratios, i.e., the total storage capacity of upstream dams divided by average annual runoff, were larger than 1. If a dam with storage capacity equaling the annual runoff is established at the outlet of the catchment without any dam before, both the reservoir index defined in this study and the storage ratio defined by FitzHugh et al. (2011) increase from 0 to 1. In this special case, the annual maximum discharges change by -21.2% (with $CI = [-29.9\%, -11.4\%]$) in this study, comparable to the 20% decrease from FitzHugh et al. (2011). It is noteworthy that this study only focuses on the effects of human factors on annual maximum discharges. Generally, the effects are larger for less frequent floods. Zhao et al. (2020) investigated floods in 1403 catchments in the US and found a decrease of 100-year floods by more than 60% in 47% of catchments with a dam upstream.

### 5.3 Insights toward a national investigation of flood changes

This study takes the first step to explain flood changes quantitatively on a national scale in China. In this study, urbanization and dam constructions significantly change annual maximum discharges in the middle and down streams of the Yellow River Basin and the Haihe River Basin, where step changes were detected by Yang et al. (2019). As a major human residence with a high density of population, the North Plain of China experiences fast urbanization in recent years (Du et al., 2018), which brings larger flood risks on lives and properties. In addition, the degree of dam regulation is larger in northern China because the annual runoff is much smaller than that in wet southern China. In this study, after removing the catchments with nonnegligible impacts of urbanization and dams, unexplained decreasing annual maximum discharges show spatial coherence in the middle and down streams of the Yellow River Basin and the upper streams of the Haihe River Basin, where decreasing trends were also derived by Yang et al. (2019). Yang et al. (2019) interpreted these trends as the results of soil conservation practices (Bai et al., 2016) and decreasing extreme rainfall (Yang et al., 2013; Wu et al., 2016). Besides, other reasons include decreasing soil moisture (Cheng et al., 2015; Yang et al., 2020a) and the impacts of cascade small soil-retaining dams (Yang et al., 2020b). It indicates that the impact factors of floods are complex in this region and further studies are required.

Caution is required to interpret the flood changes attributed to urbanization and dam constructions on a national scale because the sensitivity of floods to these factors is derived from a subset of catchments. Although the catchments used for sensitivity calculation and the ones used for flood change attribution have similar frequency distributions of urban areas and reservoir indexes in Fig. A2 and A3, these different sets of catchments may not be completely homogeneous in terms of all characteristics (topography, climate, etc.). Moreover, one should also be cautious to apply the sensitivity results to other 375 regions such as catchments in other countries.

## 6 Conclusions

We conducted a data-based analysis on the causal effects of human impacts on floods using a panel regression on a national scale, based on annual maximum discharges ($Q$) from 757 non-nested catchments in China, CCI-LC land cover data, and GRanD dam data. Specifically, we derived nationally generalizable information about the sensitivity of $Q$ to human factors, 380 namely the changes in urban areas, cropland areas, and reservoir indexes for large and medium dams. Furthermore, using a dataset of 2739 streamflow stations, we determined the explained and unexplained changes in floods by the human factors on a national scale based on the sensitivity of $Q$ to human factors. The major findings are as follows.

- Floods are sensitive to the changes in urban areas and dams. Urban areas have a positive and stable effect on floods, i.e., a 1% point increase in urban areas causes a 3.9% increase in annual maximum discharges with a confidence interval 385 $CI = [1.9\%, 5.7\%]$. Cropland areas have no significant effect on $Q$. Reservoir index has a negative and decreasing effect on $Q$, i.e., the decrease of $Q$ caused by a 1 unit increase of reservoir indexes ranges from 21.4% (with $CI = [11.4\%, 29.9\%]$) to 6.2% (with $CI = [3.2\%, 9.1\%]$) corresponding to initial reservoir indexes from 0 to 3.

- Urbanization is not a predominant driver of the increases in flood magnitudes on a national scale. In 1992-2017, increasing urban areas cause increases in $Q$ of more than 10% in 184 (11.3%) of 1625 catchments. These catchments are 390 mainly located in the North Plain of China, especially in the Huaihe River Basin and the middle and down streams of the Haihe River Basin. However, among these 184 catchments, only 5 of them have significant increases in observed $Q$.

- Dam construction is a predominant driver of the decreases in flood magnitudes on a national scale. Among the 536 catchments with at least one dam in 1960-2017, increasing reservoir indexes cause decreases in $Q$ of more than 10% in 196 (36.6%) catchments. Spatially, the impacts of dams on floods are larger in northern basins, including the Huaihe 395 River Basin, the Haihe River Basin, the Yellow River Basin, and the Songhua and Liaohe River Basin. There are 138 of those 196 catchments having significant decreases in observed $Q$, accounting for 59.0% of the total 234 catchments with significant decreases in observed $Q$.

- Unexplained decreases in flood magnitudes show spatial coherence in the middle and down streams of the Yellow River Basin and the upper streams of the Haihe River Basin. Among 1249 catchments with less than 10% changes in $Q$ caused 400 by urban areas or dams, 403 (32.3%) catchments have significant decreases in $Q$ during 1960-2017, and 46.7% of the 403 catchments are located in the Yellow River Basin and the Haihe River Basin.

This study extends the panel regression method to quantify the effects of multiple human factors on floods, which helps understand the causes of flood changes on a national scale in China. Future studies may collect more data to consider more human factors and quantify the effects on different return periods of floods.

405  **Appendix A**

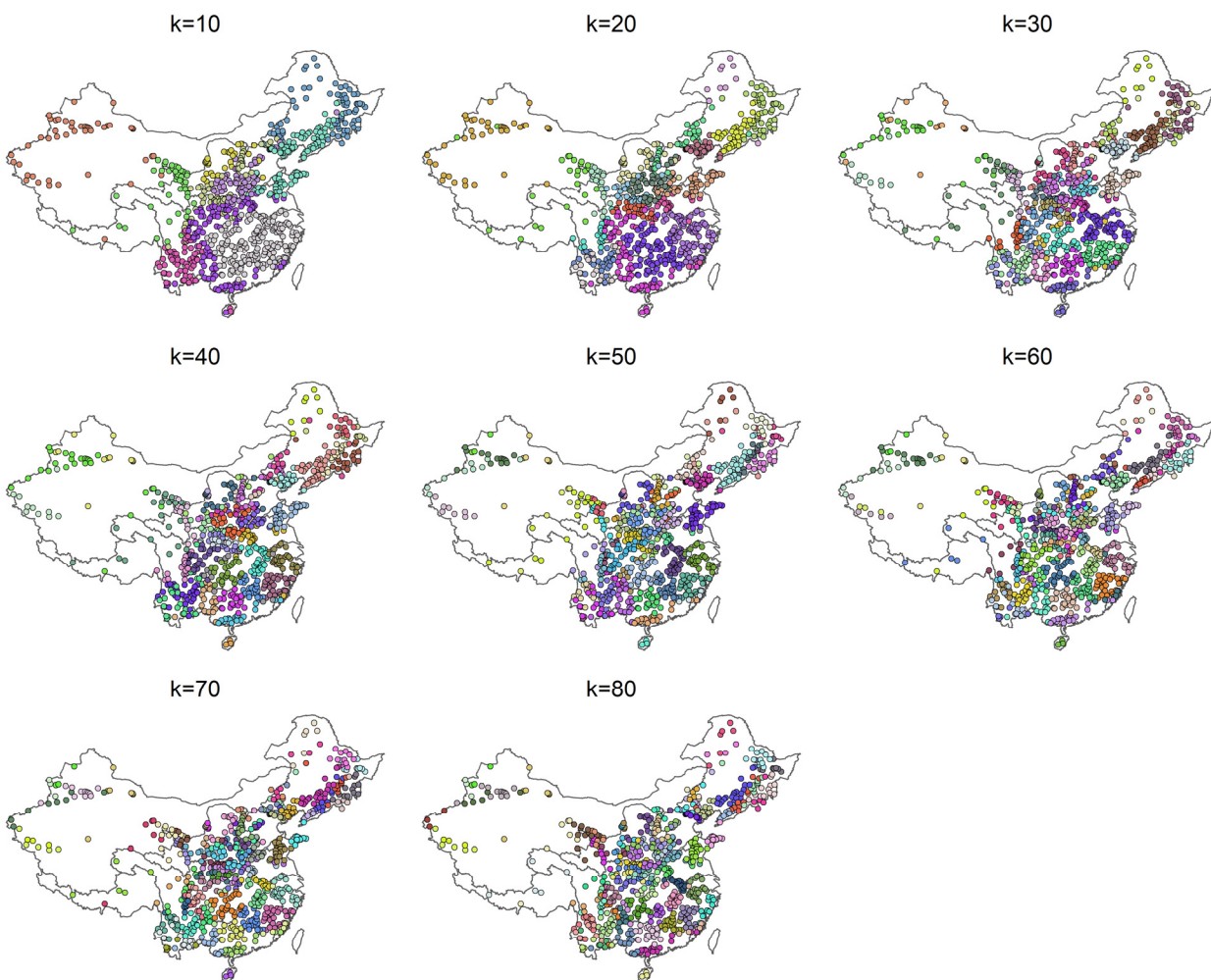

**Figure A1. Catchment groups for all $k$ values in Eq. (1). The points are the geometric centers of catchments.**

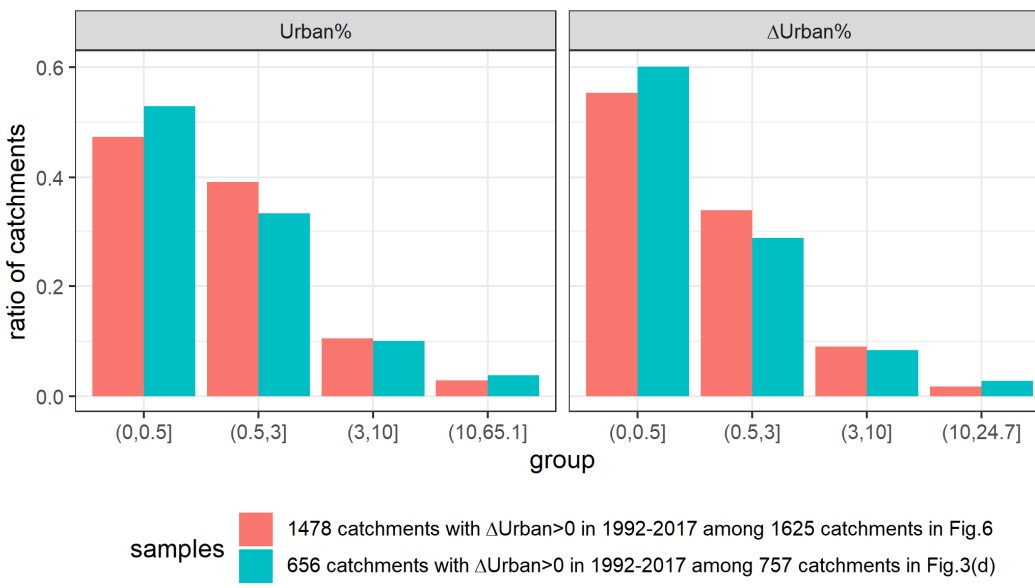

Figure A2. Frequency distribution of *Urban* and *ΔUrban* in catchments with *ΔUrban* > 0 from two catchment sets: the one used for the regression in Fig. 3(d) (green) and the one used for the flood change attribution in Fig. 6 (red).

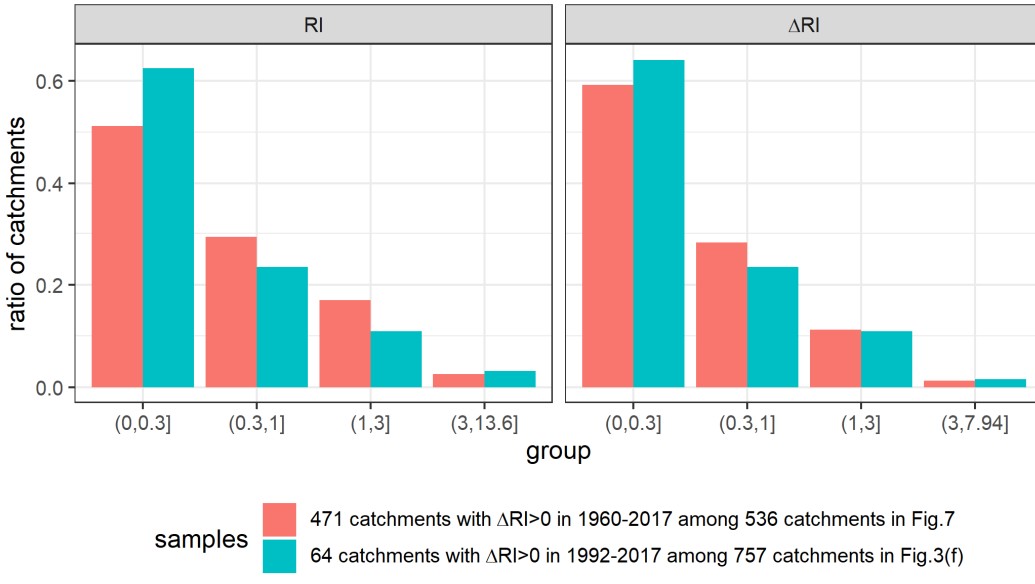

Figure A3. Frequency distribution of *RI* and *ΔRI* in catchments with *ΔRI* > 0 from two catchment sets: the one used for the regression in Fig. 3(f) (green) and the one used for the flood change attribution in Fig. 7 (red).

## Data availability

Annual maximum discharge data are obtained from the Water Resources Information Center of the Ministry of Water Resources in China (http://www.mwr.gov.cn/english/). CCI-LC data are obtained from the ESA Climate Change Initiative - Land Cover project 2017 (http://maps.elie.ucl.ac.be/CCI/viewer/download.php). GRanD data are obtained from Global Dam Watch (http://globaldamwatch.org). Köppen-Geiger climate class data and MSWEP V2.2 precipitation data are obtained from GLOH2O (http://www.gloh2o.org/).

## Author contribution

Wencong Yang: Conceptualization, Methodology, Software, Visualization, Writing – original draft preparation.

Hanbo Yang: Conceptualization, Data curation, Funding acquisition, Supervision, Writing – review & editing.

Dawen Yang: Data curation, Funding acquisition.

Aizhong Hou: Data curation.

## Competing interests

The authors declare that they have no conflict of interest.

## Financial support

This research was partially supported by funding from the National Natural Science Foundation of China (Grant Nos. 51979140 and 41661144031), the Scientific Research Project from the China Three Gorges Corporation (Grant No. 202003098), and the National Program for Support of Top-notch Young Professionals.

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
