# Peer review of "Causal effects of dams and land cover changes on flood changes in mainland China"

_Hydrology and Earth System Sciences, 2020_

## Referee Comment (RC1) · Anonymous Referee #1 · 13 Jan 2021

In the paper, the authors estimate the effects of urbanization, cropland area, and dam regulation on the magnitude of annual maximum streamflow by analyzing historical data from 2739 gaged catchments in China. The authors use panel regression methods to identify these effects, and find that urbanization increases annual maximum streamflow, whereas dam regulation decreases annual maximum streamflow. Overall, the paper addresses an important question and provides new understanding about the factors leading to changes in annual floods. The paper is well-organized and clear, with ample references to previous studies, and the methods are appropriate for the questions studied. There are a few issues that could be addressed with minor revisions, which I have detailed below. First, more detail about the underlying data for the regression is necessary to fully interpret the results. Second, I would also advise against

comparing the magnitude of p-values as a method to select between two different models. I have also noted some areas of the text needing clarification and included one suggestion for additional analysis that I think could be of interest to readers.

Main Comments

1. It would be helpful if the authors included more detail about the underlying distributions (or ranges) of the causal factors studies. For example, the authors state that the effect of urbanization is stable (i.e. linear), but it is unclear over what range of urbanization values this was tested, and if the effect could be increasing/decreasing outside of this range. Also, the panel regression specifically models within-unit (catchment) effects. This requires that there are within-catchment variations in the explanatory variables, but it is unclear how some of the variables, like dam regulation, are distributed in space and time. What percentage of catchments have changes in reservoir index, urbanization, and/or cropland area over the time period? Suggestions for ways to add this information include reporting these statistics within the text and/or adding a figure(s) of single or joint probability distributions or a time series of the causal factors.

2. Fig. 2 indicates that the lowest p-value is used to choose between a quadratic term ("increasing effect") or square root term ("decreasing effect") in cases where both terms have a p-value less than 0.01. P-values are not intended to be used to determine which model form is correct. It would be more appropriate to use a criterion designed for model selection, such as the AIC or BIC. Alternatively, in cases where both an increasing and decreasing effect are plausible and there is not a model form that clearly fits better, the authors could report both models.

3. In Fig. 8, the authors provide a map showing stations that had relatively small changes in urbanization or reservoir index but did have significant changes (>10%) in annual maximum streamflow. It would be interesting to extend this type of comparison for the stations shown in Fig. 6 and 7. In other words, it would be interesting to compare the observed streamflow trends to the trends predicted by changes in urban-

ization and RI shown in Fig 6 and 7. This comparison could identify regions where additional causal factors are involved (and thus could identify interesting areas for future research). I do not believe this analysis is necessary for publication, but would likely be of interest to readers. Comparing observed trends with trends predicted by causal factors would also be a relatively new addition to the panel regression literature within hydrology.

Line-by-line comments:

Line 31: "whether a factor affects floods?" would read more clearly as "does a factor affect floods?"

Lines 47-64: In this paragraph, I find the descriptions of the existing methodological approaches to be unclear, particularly for the first method. Is the first "model-based" method referring to an empirical model, a physically-based model, or some combination?

Line 116: "the time-varying constant effects" should just be "the constant effects"

Lines 119-121: "Although ðĺŚĚl may correlate with ðĺŚĹrban and Crop the effects of dams and land cover on floods can be derived independently since we have controlled their common drivers (Pearl and Mackenzie, 2020) in each equation, i.e., the regional time-varying term and the individual time-invariant term". I'm not sure this is correct - it is possible that RI could be temporally correlated (within-watershed) with Urban or Crop, in which case the effects could be confounded if they are modeled separately. The results don't seem to indicate that the variables are confounded, but this could also be checked by calculating the within-unit correlations.

Line 138: "Since the pooling samples were sufficient for statistical inference,..." . It is unclear what is meant by this statement, so it should be clarified.

Lines 145-147: "1. No other important..." would read more clearly as "1. There are no other important...". Likewise, "2. No interactions between..." would read more clearly

as "2. There are no interactions between. . ."

Line 155: different exponential formatting is used between Equations 6 and 7 (exp vs e). Equation 7 is also basically the same as Equation 6.

Line 158: "$|exp(g(Xi,t2) − 0) − 1| < 10\%$" . I believe the "-0" should be replaced with g(Xi,t1)?

Lines 158-160: Are the trends in annual maximum streamflow calculated using log(Q)? This is what I expect based on the presentation of the results, but should be clarified in the text.

Line 191 (and elsewhere): Suggest changing "large and middle dams" to "large and medium dams" or "medium and large dams".

Line 199: "The number of catchment groups ðÍŚŸ in Section 2.2 had no optimal value." Was there a method used to test for an optimal value?

Line 209: "with only one exceptional type of effect for Urban and Crop". This phrasing is confusing, so I suggest rephrasing to something like: "except for two cases (Urban effect when k=10, and Crop effect when k=40)".

Line 210: "percentage increase" – I believe this should be a percentage point increase? (For example, an increase from 10% to 11% is a 10 percent increase, or a 1 percentage point increase.)

Line 238 and 348: "more than 10% of increases in Q". If I understand correctly, this should be written as "increases in Q of more than 10%".

---

## Referee Comment (RC2) · Marc F. Muller (Referee) · 21 Jan 2021

The study applies a panel research design to estimate the causal effect of three hypothesized human-related drivers (urban extent, cropland extent and reservoir regulation) of annual flood peaks in China. While the methodological contributions of the study are (in my view) limited compared to recent other studies using panel regressions in a similar context (e.g., Blum 2020 and Davenport 2020 cited in the study), the study is nonetheless valuable in that it provides important insights on how these process operate in conjunction, using a very large dataset in China. The study is in my view appropriate for publication in HESS, provided the author address the following major concerns that I have.

[Figure]

1. Potentially misleading map figures. To be clear, the panel approach does *not* allow to estimate heterogeneous treatment effects. It allows to estimate one average effect of (say) urban expansion on flow peaks (i.e. one single value of beta, if g() is linear) across the whole sample. It does *not* allow to say that urban expansion has a larger effect on flood peaks in some regions than in others. Yet the maps in figures 6 and 7 (and their discussion throughout the paper) appear to suggest exactly that, which I find misleading. The spatial variability in the "effect" of crop/urban on floods represented in these maps only emerges because changes crop and urban cover are themselves varying across regions. Figure 6 is nothing more than a map of urban cover change, scaled by a constant factor (the estimated beta) representing the linear effect it has on flood peaks. This point is important to clarify throughout the text, at the very least by specifying the estimated value of beta and theta in the captions of Figures 6 and 7 (see minor comments for other suggestions).

2. Fixed Effects. I am wondering why you use "regions" as space fixed effects, and not the individual basins themselves. For Blum et al., this approach made sense because they interact the treatment (X) with covariates (e.g., soil permeability, etc) but I don't really see the point of doing that here. I am concerned that it might introduce a bias associated with varying confounding factors within the regions (e.g., basin altitude can vary within regions and affect both the treatment — crop or urban cover — and flood magnitude). Adding a specification with basin-level fixed effect (i.e. setting k==number of basins) as robustness check might help alleviate my concern.

3. Heterogeneous treatment effect: I am wondering if your results are affected by heterogenous treatment effects in the sense that most basins of the sample likely have little impervious surface cover. (By the way, please add a table with descriptive statistics for the reader to assess that). If the deviates (even slightly) from the three arbitrary functional forms that you impute to g(), this may potentially bias your average estimates. A way to control for this (perhaps) would be to do a robustness check by running the analysis to a subset of highly (lowly) impervious basin to see how sensitive the effect

is.

4. Nestedness: Finally, the ordinary least square estimator that (I assume) you are using only provides an unbiased estimate of standard errors if residuals (epsilon) are independant. In your case, I am concerned that many of your observations might be nested (i.e. taken along different reaches of a same river), which might introduce a correlation in the epsilon. For instance a time- and space- specific shock on flow peaks observed in a headwater catchment will likely affect flow peaks observed at several gauges along that river. The fact that errors congregate around specific basins in Figure 8 is actually a strong indication of that effect! This effect might lead you to underestimate the standard errors on your regression coefficient and find a significant effect where there is none. A way to address that would be to use the topology of your river network to specify the structure of your variance-covariance matrix (see, e.g., Muller and Thompson 2015) which you can then incorporate in your estimation via Generalized Least Square or Restricted Maximum Likelihood. Alternatively, you could do a robustness check where you run your OLS estimation on multiple subsets of your full sample, for which you made sure that all observations are from different catchments — hopefully the results will be similar.

Minor Comments.

The first sentence of the abstract is awkward ("because the knowledge and observations toward the effects are limited"). Please reformulate.

L79: middle—> medium ?

L94: It took me a while to realize that you *defined* your regions such that climate is homogeneous within them (as oppose to assuming that climate is homogeneous within a bunch of predetermined regions). Maybe clarify that here?

Eqn 6: My understanding is that $\Delta Q$ varies of space but not time: if so, how to you "average over" the time index in the middle expression. Also, this would be an ideal place

to clarify that ∆Q varies in space only because ∆X varies in space. Your estimation of g() is constant in space and time.

Fig 2: I agree with the other reviewer that p-values are an odd criteria for model selection. Either justify it, or use goodness of fit metric.

L155-160 and Fig 8. I find it a good idea to analyze the spatial distribution of model deviations (i.e. locations where variations in Q are not explained by the modeled drivers), but I find the approach chosen to identify these locations odd/arbitrary and challenging to understand. Wouldn't it be more straightforward to simply map the temporal variance of the residuals (i.e. $Var\_i(Eps\_it)$)?

L294 "Coefficient of Variation" can be understood as the ratio between the standard deviation and the mean. I don't think that's what you mean here, so please reformulate.

L294. You provide a good illustrative example of the models inability to capture heterogeneous treatment in time, but here would also be a good opportunity to give an example of a heterogeneous treatment in space (i.e. a scenario where cropland might persistently have a stronger effect on flow peaks in some locations than in other ). That would contribute alleviating my first major concern, above.

SI. Please add a descriptive statistics table with key stats on all the considered variable across your sample.

Marc Muller

References Blum, A. Et al (2020) Causal Effect of Impervious Cover on Annual Flood Magnitude in the United States, GRL

Davenport et al. 2020 Flood Size Increases Nonlinearly Across the Western United States in Response to Lower Snow-Precipitation Ratios, WRR

Muller, M.F. and Thompson, S.E. (2015) "TopREML: a topological restricted maximum likelihood approach to regionalize trended runoff signatures in stream networks", HESS

---

## Author Response (AR1)

Dear Editor and Reviewers,

Thank you very much for the assessment of our manuscript. We appreciate that you provide an opportunity for us to improve our study. We have revised our manuscript thoroughly according to the comments. Please see the replies to the reviewers' comments below. In the response, the blue texts are the comments and the green texts are the quotes of the manuscript.

On behalf of all co-authors,

Wencong Yang

**Reviewer #1 Comment 1:** (hereafter referred to as R1C1, R1C2…) *In the paper, the authors estimate the effects of urbanization, cropland area, and dam regulation on the magnitude of annual maximum streamflow by analyzing historical data from 2739 gaged catchments in China. The authors use panel regression methods to identify these effects, and find that urbanization increases annual maximum streamflow, whereas dam regulation decreases annual maximum streamflow. Overall, the paper addresses an important question and provides new understanding about the factors leading to changes in annual floods. The paper is well-organized and clear, with ample references to previous studies, and the methods are appropriate for the questions studied. There are a few issues that could be addressed with minor revisions, which I have detailed below. First, more detail about the underlying data for the regression is necessary to fully interpret the results. Second, I would also advise against comparing the magnitude of p-values as a method to select between two different models. I have also noted some areas of the text needing clarification and included one suggestion for additional analysis that I think could be of interest to readers.*

**A:** Thank you for your constructive comments. We have carefully considered your suggestions and revised the paper. We have made the following major revisions in the method and data.

First, we added 3-day and 30-day total precipitation before flood peaks for each catchment in the regression to account for individual time-varying confounders. The reason for such a revision is that the delineated climate regions cannot fully control climatic confounders since the climatic drivers of floods have sub-regional spatial variability. Therefore, the regression equation has been revised as (please see Eq. (1) in the revised manuscript):

$$log(Q_{i,t}) = \alpha_i + g_1\left(Urban_{i,t}\right) + g_2\left(Crop_{i,t}\right) + g_3\left(RI_{i,t}\right) + \pi_{r,t}D_rD_t$$

$$+ D_r\left(\varphi_r P_{i,t}^{(3)} + \lambda_r P_{i,t}^{(30)}\right) + \varepsilon_{i,t}$$

where $P_{i,t}^{(3)}$ is the 3-day total precipitation before the flood peak in year $t$ of catchment $i$, which accounts for the rainfall that causes the flood; $P_{i,t}^{(30)}$ is the 30-day total precipitation before the flood peak in year $t$ of catchment $i$, which accounts for the soil moisture and snowmelt that cause the flood. The coefficients of $P_{i,t}^{(3)}$ and $P_{i,t}^{(30)}$, namely $\varphi_r$ and $\lambda_r$, are assumed to be constant within a climatic region $r$. The original region term $\pi_{r,t}D_rD_t$ accounts for omitted time-varying regional confounders other than $P_{i,t}^{(3)}$ and $P_{i,t}^{(30)}$.

Second, we selected 757 non-nested catchments to fit the regression model so that the residuals of the model were not highly correlated. This revision avoids uncorrected inference about the regression coefficients due to the underestimation of their standard deviations using correlated flood samples.

The results did not substantially change after the methodology revision above. Other issues you mentioned are addressed in the following replies.

*Main Comments*

**R1C2:** *1. It would be helpful if the authors included more detail about the underlying distributions (or ranges) of the causal factors studies. For example, the authors state that the effect of urbanization is stable (i.e. linear), but it is unclear over what range of urbanization values this was tested, and if the effect could be increasing/decreasing outside of this range. Also, the panel regression specifically models within-unit (catchment) effects. This requires that there are within-catchment variations in the explanatory variables, but it is unclear how some of the variables, like dam regulation, are distributed in space and time. What percentage of catchments have changes in reservoir index, urbanization, and/or cropland area over the time period? Suggestions for ways to add this information include reporting these statistics within the text and/or adding a figure(s) of single or joint probability distributions or a time series of the causal factors.*

**A:** Thank you very much for your critical comment. We agree with you that a statistical summary of catchment characteristics is necessary. We have added the summary table (Table 1) in the revised manuscript.

**Table 1. Summary of catchment characteristics for 757 independent catchments. For each catchment, among its all years with available flood data in 1992-2017, we choose the last year to calculate *Urban*, *Crop*, and *RI*, and choose the first and last year to calculate Δ*Urban*, Δ*Crop*, and Δ*RI*. The summaries of *RI* and Δ*RI* are calculated based on 207 catchments with at least one large and medium dam.**

| Variables | Min. | 1st Qu. | Median | Mean | 3rd Qu. | Max. |
|---|---|---|---|---|---|---|
| *Area* ($km^2$) | 29 | 499 | 1096 | 3341 | 2763 | 142372 |
| *Urban* (%) | 0 | 0.06 | 0.30 | 1.52 | 1.10 | 65.07 |
| Δ*Urban* (%) | 0 | 0.05 | 0.23 | 1.14 | 0.85 | 24.66 |
| *Crop* (%) | 0 | 10.63 | 24.71 | 32.75 | 48.99 | 99.58 |
| Δ*Crop* (%) | -21.58 | -0.81 | -0.02 | 0.38 | 0.87 | 32.04 |
| *RI* | 0.01 | 0.09 | 0.21 | 0.51 | 0.61 | 7.45 |
| Δ*RI* | 0 | 0 | 0 | 0.17 | 0.07 | 7.44 |

**R1C3:** *2. Fig. 2 indicates that the lowest p-value is used to choose between a quadratic term ("increasing effect") or square root term ("decreasing effect") in cases where both terms have a p-value less than 0.01. P-values are not intended to be used to determine which model form is correct. It would be more appropriate to use a criterion designed for model selection, such as the AIC or BIC. Alternatively, in cases where both an increasing and decreasing effect are plausible and there is not a model form that clearly fits better, the authors could report both models.*

**A:** Thank you for your important comment. We agree with you that p-values are not appropriate for choosing models. In the revision, we used AIC to select the optimal model from all 27 possible combinations of $[g_1(\cdot), g_2(\cdot), g_3(\cdot)]$ types. This new model selection criterion did not change the optimal model form compared with the original manuscript, i.e., $g_1(X_{i,t}) = \beta X_{i,t}$, $g_2(X_{i,t})$ is not statistically significant, and $g_3(X_{i,t}) = \theta X_{i,t}^{1/2}$. In the revised manuscript, we deleted the original Figure 2 and added the use of AIC in Line 137-138 as "To determine the specific effect type, we fitted regressions with 27 possible combinations of $[g_1(\cdot), g_2(\cdot), g_3(\cdot)]$ types and selected the one with the lowest AIC value."

**R1C4:** *3. In Fig. 8, the authors provide a map showing stations that had relatively small changes in urbanization or reservoir index but did have significant changes (>10%) in annual maximum streamflow. It would be interesting to extend this type of*

*comparison for the stations shown in Fig. 6 and 7. In other words, it would be interesting to compare the observed streamflow trends to the trends predicted by changes in urbanization and RI shown in Fig 6 and 7. This comparison could identify regions where additional causal factors are involved (and thus could identify interesting areas for future research). I do not believe this analysis is necessary for publication, but would likely be of interest to readers. Comparing observed trends with trends predicted by causal factors would also be a relatively new addition to the panel regression literature within hydrology*

**A:** Thank you very much for your valuable comment. You provided an interesting idea to compare observed flood trends with the trends predicted by the changes in *Urban* and *RI*. However, in addition to gradual changes (trends), observed floods may also experience abrupt changes. It is difficult to determine the percentage change of floods between two specific years using observed flood data with abrupt changes, especially using data in a short period (26 years from 1992 to 2017 in this study). It is beyond the scope of this study to derive the percentage change of floods under different kinds of flood change patterns. Nonetheless, we still hope to see whether catchments with significant changes in observed floods match the ones with large changes in *Urban* and *RI*. Therefore, in the revised manuscript, we have labeled the catchments with significant changes in observed floods according to the Mann-Kendall test and Pettitt's test in Fig. 6 and Fig. 7.

*Line-by-line comments:*

**R1C5:** *Line 31: "whether a factor affects floods?" would read more clearly as "does a factor affect floods?"*

**A:** Thank you very much for your suggestion. We have changed the sentence in the revised manuscript as "1. does a factor affect floods?" (Please see Line 32-33).

**R1C6:** *Lines 47-64: In this paragraph, I find the descriptions of the existing methodological approaches to be unclear, particularly for the first method. Is the first "model-based" method referring to an empirical model, a physically-based model, or some combination?*

**A:** Thank you very much for your comment. The first method refers to the use of physical hydrological models, which can simulate floods in different scenarios (e.g., with and without human impacts). We have changed the "model-based approach" to "physical model simulation" in the revised manuscript (Please see Line 48-49).

**R1C7:** *Line 116: "the time-varying constant effects" should just be "the constant effects"*

**A:** Thank you very much. We have changed the sentence to "$\pi_{r,t}$ is the constant effects of region $r$ in year $t$." (Please see Line 120)

**R1C8:** *Lines 119-121: "Although RI may correlate with Urban and Crop, the effects of dams and land cover on floods can be derived independently since we have controlled their common drivers (Pearl and Mackenzie, 2020) in each equation, i.e., the regional time-varying term and the individual time-invariant term". I'm not sure this is correct - it is possible that RI could be temporally correlated (within-watershed) with Urban or Crop, in which case the effects could be confounded if they are modeled separately. The results don't seem to indicate that the variables are confounded, but this could also be checked by calculating the within-unit correlations.*

**A:** Thank you very much for your comment. *RI* and *Urban* or *Crop* have step changes, so with-unit correlations cannot capture their dependence relationship. In the revision, we combined *Urban*, *Crop*, and *RI* into one regression model (please see Eq. (1) in the revised manuscript). Note that the effects of *Urban*, *Crop*, and *RI* on floods have little change after revising the regression model.

**R1C9:** *Line 138: "Since the pooling samples were sufficient for statistical inference,…". It is unclear what is meant by this statement, so it should be clarified.*

**A:** We no longer used p values to choose model forms (please see R1C3). Therefore, this sentence has been deleted from the revised manuscript.

**R1C10:** *Lines 145-147: "1. No other important: …" would read more clearly as "1. There are no other important…". Likewise, "2. No interactions between…" would read more clearly as "2. There are no interactions between…"*

**A:** Thank you very much. In the revised manuscript, we have changed the sentences to "1. There are no other important time-varying sub-regional variables that significantly affect both human factors and floods; 2. There are no interactions between human factors and regional or individual characteristics that produce significant spatially heterogeneous effects." (Please see Line 141-143)

**R1C11:** *Line 155: different exponential formatting is used between Equations 6 and 7 (exp vs e). Equation 7 is also basically the same as Equation 6.*

**A:** Thank you very much. We have unified the exponential formatting by using "*exp*" in the revised manuscript. The original Equations 6 (Equation 5 in the revised manuscript) shows the sensitivity of $Q$ to $X$, i.e., "the percentage change in $Q$ given a fixed change in $X$", as described in Line 132. The fixed change, $\Delta X$, equals 1% for $X$=*Urban* or *Crop* and 1 unit for $X$=*RI*. While the original Equation 7 (Equation 6 in the revised manuscript) shows the accumulated changes in $Q$ due to the changes of $X$ in a long period, i.e., "the accumulated flood changes attributed to the change in factor $X$ for catchment $i$ from year $t_1$ to $t_2$", as described in Line 148.

**R1C12:** *Line 158: "|exp(g(Xi,t2) – 0) – 1| < 10%" . I believe the "-0" should be replaced with g(Xi,t1)?*

**A:** Thank you very much for your comment. It should be "0". Here we hope to select catchments that are free from the impacts of urbanization and dam constructions. Therefore, the initial value of the factor $(X_{i,t_1})$ equals 0, which represents the condition with no urban areas and no dams. In the revised manuscript, we have changed the sentences to "To examine how floods changed in catchments that were free from the impacts of urban areas, cropland areas, and dams, we selected catchments with less than 10% changes in flood peaks due to those factors respectively. Specifically, for a factor $X$, we selected catchments with $\left| \exp\left(g\left(X_{i,t_2}\right) - 0\right) - 1 \right| < 10\%$ where $t_2$ was the most recent year of the data." (Please see Line 150-152)

**R1C13:** *Lines 158-160: Are the trends in annual maximum streamflow calculated using log(Q)? This is what I expect based on the presentation of the results, but should be clarified in the text.*

**A:** Thank you very much. The trends in annual maximum streamflow are calculated using $Q$. In the revised manuscript, we have changed the sentence to "Then, we applied the Mann-Kendall test (Mann, 1945) and Pettitt's test (Pettitt, 1979) in those catchments to detect the ones with significant changes in $Q$." (Please see Line 152-154)

**R1C14:** *Line 191 (and elsewhere): Suggest changing "large and middle dams" to "large and medium dams" or "medium and large dams".*

**A:** Thank you very much for your suggestion. We have changed all "large and middle dams" to "large and medium dams" in the revised manuscript.

**R1C15:** *Line 199: "The number of catchment groups k in Section 2.2 had no optimal value." 'Was there a method used to test for an optimal value?*

**A:** Thank you very much for your comment. Indeed, we can use the Silhouette value of the clustering or the AIC value of the regression model to select a $k$ value. However, these methods derive an optical $k$ from a statistical perspective rather than from a hydrological perspective. We prefer to test the sensitivity of the model to varying $k$ so that we can test the robustness of our method.

**R1C16:** *Line 209: "with only one exceptional type of effect for Urban and Crop". This phrasing is confusing, so I suggest rephrasing to something like: "except for two cases (Urban effect when k=10, and Crop effect when k=40)".*

**A:** Thank you very much for your suggestion. In the revised manuscript, the effect types of *Urban* and *Crop* are consistent for varying values of $k$, and thus, we have removed this sentence.

**R1C17:** *Line 210: "percentage increase" – I believe this should be a percentage point increase? (For example, an increase from 10% to 11% is a 10 percent increase, or a 1 percentage point increase.)*

**A:** Thank you very much for pointing out the typo. Yes, the "percentage increase" should be "percentage point increase". We have corrected the word in the revised manuscript.

**R1C18:** *Line 238 and 348: "more than 10% of increases in Q". If I understand correctly, this should be written as "increases in Q of more than 10%".*

**A:** Thank you very much. In the revised manuscript, we have revised the sentences as "Increasing *Urban* causes increases in $Q$ of more than 10% in 184 (11.3% of 1625) catchments" (please see Line 241), and "increasing urban areas cause increases in $Q$ of more than 10% in 184 (11.3%) of 1625 catchments" (please see Line 372).

**Reviewer #2 Comment 1:** (hereafter referred to as R2C1, R2C2…) *The study applies a panel research design to estimate the causal effect of three hypothesized human-related drivers (urban extent, cropland extent and reservoir regulation) of annual flood peaks in China. While the methodological contributions of the study are (in my view) limited compared to recent other studies using panel regressions in a similar context (e.g., Blum 2020 and Davenport 2020 cited in the study), the study is nonetheless valuable in that it provides important insights on how these process operate in conjunction, using a very large dataset in China. The study is in my view appropriate for publication in HESS, provided the author address the following major concerns that I have*

**A:** Thank you for your constructive comments. We have carefully considered your suggestions and addressed your concerns. We have made the following major revisions in the method and data.

First, we added 3-day and 30-day total precipitation before flood peaks for each catchment in the regression to account for individual time-varying confounders. The reason for such a revision is that the delineated climate regions cannot fully control climatic confounders since the climatic drivers of floods have sub-regional spatial variability. Therefore, the regression equation has been revised as (please see Eq. (1) in the revised manuscript):

$$log(Q_{i,t}) = \alpha_i + g_1\left(Urban_{i,t}\right) + g_2\left(Crop_{i,t}\right) + g_3\left(RI_{i,t}\right) + \pi_{r,t}D_rD_t$$

$$+ D_r\left(\varphi_r P_{i,t}^{(3)} + \lambda_r P_{i,t}^{(30)}\right) + \varepsilon_{i,t}$$

where $P_{i,t}^{(3)}$ is the 3-day total precipitation before the flood peak in year $t$ of catchment $i$, which accounts for the rainfall that causes the flood; $P_{i,t}^{(30)}$ is the 30-day total precipitation before the flood peak in year $t$ of catchment $i$, which accounts for the soil moisture and snowmelt that cause the flood. The coefficients of $P_{i,t}^{(3)}$ and $P_{i,t}^{(30)}$, namely $\varphi_r$ and $\lambda_r$, are assumed to be constant within a climatic region $r$. The original region term $\pi_{r,t}D_rD_t$ accounts for omitted time-varying regional confounders other than $P_{i,t}^{(3)}$ and $P_{i,t}^{(30)}$.

Second, we selected 757 non-nested catchments to fit the regression model so that the residuals of the model were not highly correlated. This revision avoids uncorrected inference about the regression coefficients due to the underestimation of their standard deviations using correlated flood samples.

Note that the results do not substantially change after the methodology revision above. While we believe the revised method and data are more convincing and solid.

**R2C2:** *1. Potentially misleading map figures. To be clear, the panel approach does \*not\* allow to estimate heterogeneous treatment effects. It allows to estimate one average effect of (say) urban expansion on flow peaks (i.e. one single value of beta, if g() is linear) across the whole sample. It does \*not\* allow to say that urban expansion has a larger effect on flood peaks in some regions than in others. Yet the maps in figures 6 and 7 (and their discussion throughout the paper) appear to suggest exactly that, which I find misleading. The spatial variability in the "effect" of crop/urban on floods represented in these maps only emerges because changes crop and urban cover are themselves varying across regions. Figure 6 is nothing more than a map of urban cover change, scaled by a constant factor (the estimated beta) representing the linear effect it has on flood peaks. This point is important to clarify throughout the text, at the very least by specifying the estimated value of beta and theta in the captions of Figures 6 and 7 (see minor comments for other suggestions).*

**A:** Thank you very much for your critical comment. As we know, the panel regression in this paper is unable to estimate heterogeneous effects across different catchments. The "average effect" of a factor on floods you mentioned is shown in Fig. 4 and Fig. 5, where the homogeneous sensitivities of floods to human factors across all catchments are presented. Fig. 6 and Fig. 7 show the accumulated changes in floods due to the changes in human factors in a long period for each catchment rather than the heterogeneous effects. To avoid misunderstanding, in the captions of Fig. 6 and Fig. 7 in the revised manuscript, we have emphasized the changes in floods as the "accumulated changes" and clarify that the changes are calculated by Eq. (6), i.e., $\Delta Q(\%) = \exp(g(X_{i,t_2}) - g(X_{i,t_1})) - 1$. The first sentences in the captions of Fig. 6 and Fig. 7 have been changed to "Accumulated increases in annual maximum discharges ($Q$) due to the increases in urban areas ($Urban$) for 1625 catchments with at least 20 years of flood data in 1992-2017, according to Eq. (6)." and "Accumulated decreases in annual maximum discharges ($Q$) due to the increases in reservoir index ($RI$) for 536 catchments with at least 30 years of flood data and at least one dam in 1960-2017, according to Eq. (6)."

**R2C3:** *2. Fixed Effects. I am wondering why you use "regions" as space fixed effects, and not the individual basins themselves. For Blum et al., this approach made sense*

*because they interact the treatment (X) with covariates (e.g., soil permeability, etc) but I don't really see the point of doing that here. I am concerned that it might introduce a bias associated with varying confounding factors within the regions (e.g., basin altitude can vary within regions and affect both the treatment crop or urban cover and flood magnitude). Adding a specification with basin-level fixed effect (i.e. setting k==number of basins) as robustness check might help alleviate my concern.*

**A:** Thank you for your important comment. We use "regions" to account for omitted time-varying confounders such as vegetation changes. If we set $k$=number of basins, i.e., we use a basin-level dummy variable interacted with a year dummy to represent time-varying confounders, the number of regression coefficients will be larger than the number of flood peak observations, which makes the estimation of coefficients infeasible. Therefore, we use a region-level dummy variable interacted with a year dummy (i.e., $\pi_{r,t} D_r D_t$) to control omitted time-varying effects. As for basin-level confounders, the time-varying effect has been controlled by the event precipitation ($P_{i,t}^{(3)}$ and $P_{i,t}^{(30)}$, see R2C1), and the time-invariant effect has been controlled by the individual-specific intercept ($\alpha_i$). Basin altitude, the sub-regional confounder you mentioned, has been included in $\alpha_i$.

**R2C4:** *3. Heterogeneous treatment effect: I am wondering if your results are affected by heterogenous treatment effects in the sense that most basins of the sample likely have little impervious surface cover. (By the way, please add a table with descriptive statistics for the reader to assess that). If the deviates (even slightly) from the three arbitrary functional forms that you impute to g(), this may potentially bias your average estimates. A way to control for this (perhaps) would be to do a robustness check by running the analysis to a subset of highly (lowly) impervious basin to see how sensitive the effect is.*

**A:** Thank you very much for your valuable comment. We agree with you that the results depend on the selection of basins. However, we believe the difference in the results brought by basin selection is more related to sampling uncertainties rather than heterogeneous treatment effects. If we manually select a subset of basins with low impervious areas to fit the model, we may get a model with a low signal-to-noise level since the changes in floods caused by changing impervious areas are far small than the model errors. Therefore, in the revision, we calculated the confidence intervals of the regression coefficients by bootstrapping, which resampled all pooling flood samples to fit the model 1000 times. Using bootstrapping, the confidence intervals of the

regression coefficients accounted for the sampling uncertainties related to basin selection and year selection. In the revised manuscript, we have added the use of bootstrapping as "We used bootstrapping to test the significance and derive the confidence intervals of coefficients $\beta$, $\gamma$, and $\theta$ so that the model residuals were allowed to be non-Gaussian and the sampling uncertainty could be accounted for" (Please Line 138-140). We present the new results of the significance tests and confidence intervals of regression coefficients in Table 2.

In addition, we have added the summary table (Table 1) in the revised manuscript.

**Table 1. Summary of catchment characteristics for 757 independent catchments. For each catchment, among its all years with available flood data in 1992-2017, we choose the last year to calculate *Urban*, *Crop*, and *RI*, and choose the first and last year to calculate ΔUrban, ΔCrop, and ΔRI. The summaries of *RI* and ΔRI are calculated based on 207 catchments with at least one large and medium dam.**

| Variables | Min. | 1st Qu. | Median | Mean | 3rd Qu. | Max. |
|---|---|---|---|---|---|---|
| *Area* ($km^2$) | 29 | 499 | 1096 | 3341 | 2763 | 142372 |
| *Urban* (%) | 0 | 0.06 | 0.30 | 1.52 | 1.10 | 65.07 |
| ΔUrban (%) | 0 | 0.05 | 0.23 | 1.14 | 0.85 | 24.66 |
| *Crop* (%) | 0 | 10.63 | 24.71 | 32.75 | 48.99 | 99.58 |
| ΔCrop (%) | -21.58 | -0.81 | -0.02 | 0.38 | 0.87 | 32.04 |
| *RI* | 0.01 | 0.09 | 0.21 | 0.51 | 0.61 | 7.45 |
| ΔRI | 0 | 0 | 0 | 0.17 | 0.07 | 7.44 |

**R2C5:** *4. Nestedness: Finally, the ordinary least square estimator that (I assume) you are using only provides an unbiased estimate of standard errors if residuals (epsilon) are independant. In your case, I am concerned that many of your observations might be nested (i.e. taken along different reaches of a same river), which might introduce a correlation in the epsilon. For instance a time- and space- specific shock on flow peaks observed in a headwater catchment will likely affect flow peaks observed at several gauges along that river. The fact that errors congregate around specific basins in Figure 8 is actually a strong indication of that effect! This effect might lead you to underestimate the standard errors on your regression coefficient and find a significant effect where there is none. A way to address that would be to use the topology of your river network to specify the structure of your variance-covariance matrix (see, e.g., Muller and Thompson 2015) which you can then incorporate in your estimation via Generalized Least Square or Restricted Maximum Likelihood. Alternatively, you could*

*do a robustness check where you run your OLS estimation on multiple subsets of your full sample, for which you made sure that all observations are from different catchments  T hopefully the results will be similar.*

**A:** Thank you very much for your comment. We indeed used an ordinary least square estimator to fit the model. So we agree with you that nested catchments cause dependence between model residuals, and thus produce wrong inference about the regression coefficients. In order to select non-nested catchments and include as many catchments with dams as possible, in the revision, we selected the most upstream catchments with large or medium dams (if possible) among overlapping catchments. We got 757 catchments from this selection, among which 207 catchments had as least one dam. Although the results did not change substantially using the new subset of catchments, the regression model in the revised manuscript was based on these 757 catchments. In the revised manuscript, we have described the catchment selection as "To avoid inaccurate statistical inference on the regression coefficients due to the correlated model residuals caused by nested catchments, we selected the most upstream catchments with large or medium dams (if possible) among overlapping catchments" (Please see Line 191-193). The spatial distribution of the selected catchments is shown in the new Fig. 3.

*Minor Comments.*

**R2C6:** *The first sentence of the abstract is awkward ("because the knowledge and observations toward the effects are limited"). Please reformulate.*

**A:** Thank you very much for your comment. We have revised the sentence as "Quantifying the effects of human activities on floods is challenging because of limited knowledge and observations" (please see Line 8-9) in the revised manuscript.

**R2C7:** *L79: middle -> medium ?*

**A:** Thank you very much for your suggestion. We have changed all "large and middle dams" to "large and medium dams" in the revised manuscript.

**R2C8:** *L94: It took me a while to realize that you \*defined\* your regions such that climate is homogeneous within them (as oppose to assuming that climate is homogeneous within a bunch of predetermined regions). Maybe clarify that here?*

**A:** Thank you very much for your comment. In the revised manuscript, we have introduced how we defined regions as "We delineate regions by climate since the

climate is the first-order driver of catchment similarity (Jehn et al., 2020). In this way, we can control the effect of many omitted variables that have spatial homogeneity." (Please see Line 98-99)

**R2C9:** *Eqn 6: My understanding is that ΔQ varies of space but not time: if so, how to you "average over" the time index in the middle expression. Also, this would be an ideal place to clarify that ΔQ varies in space only because ΔX varies in space. Your estimation of g() is constant in space and time.*

**A:** Thank you very much for your comment. The $\Delta Q$ here is the effect is the sensitivity of $Q$ to $X$ rather than the accumulated change of floods along time. As we said in Line 132, this sensitivity is "the percentage change in $Q$ given a fixed change in $X$". When we exhibited this sensitivity, we kept $X$ and $\Delta X$ to be constant across all catchments (please see Fig. 4 and Fig. 5), so the $\Delta Q$ here was constant not only in time but also in space. To avoid misunderstanding, we have changed the original Equation 6 to (please see the new Equation 5 in the revised manuscript)

$$\Delta Q(\%) = \Delta Q/Q = \exp(g(X + \Delta X) - g(X)) - 1$$

**R2C10:** *Fig 2: I agree with the other reviewer that p-values are an odd criteria for model selection. Either justify it, or use goodness of fit metric.*

**A:** Thank you very much for your comment. We agree with you that p-values are not appropriate for choosing models. In the revision, we used AIC to select the optimal model from all 27 possible combinations of $[g_1(\cdot),\ g_2(\cdot),\ g_3(\cdot)]$ types. This new model selection criterion did not change the optimal model form compared with the original manuscript, i.e., $g_1(X_{i,t}) = \beta X_{i,t}$, $g_2(X_{i,t})$ is not statistically significant, and $g_3(X_{i,t}) = \theta X_{i,t}^{1/2}$. In the revised manuscript, we deleted the original Figure 2 and added the use of AIC in Line 137-138 as "To determine the specific effect type, we fitted regressions with 27 possible combinations of $[g_1(\cdot),\ g_2(\cdot),\ g_3(\cdot)]$ types and selected the one with the lowest AIC value."

**R2C11:** *L155-160 and Fig 8. I find it a good idea to analyze the spatial distribution of model deviations (i.e. locations where variations in Q are not explained by the modeled drivers), but I find the approach chosen to identify these locations odd/arbitrary and challenging to understand. Wouldn't it be more straightforward to simply map the temporal variance of the residuals (i.e. Var_i(Eps_it))?*

**A:** Thank you very much for your suggestion. However, our purpose here is to see how floods change in catchments that are free from the impacts of urbanization and dam constructions. Therefore, we selected catchments with low *Urban* and *RI* to derive flood trends. Analyzing the spatial distribution of model deviations (e.g., to calculate $\text{Var}(\varepsilon_{i,t})$) only tells us the places where some drivers are missing in the regression. Model deviations do not tell us the directions and magnitudes of changes in floods. We have changed the original paragraph in Line 155-160 to "To examine how floods changed in catchments that were free from the impacts of urban areas, cropland areas, and dams, we selected catchments with less than 10% changes in flood peaks due to those factors respectively. Specifically, for a factor $X$, we selected catchments with $\left|\exp\left(g\left(X_{i,t_2}\right)-0\right)-1\right| < 10\%$ where $t_2$ was the most recent year of the data. Then, we applied the Mann-Kendall test (Mann, 1945) and Pettitt's test (Pettitt, 1979) in those catchments to detect the ones with significant changes in $Q$" (Please Line 150-154). We have also changed the first two sentences in the caption of Figure 8 to "Change directions of annual maximum discharges ($Q$) during 1960-2017 for 1249 catchments with at least 30 years of flood data, $Urban < 2.6\%$ and $RI < 0.19$. These catchments are considered to be free from the impacts of urbanization and dam constructions."

**R2C12:** *L294 "Coefficient of Variation" can be understood as the ratio between the standard deviation and the mean. I don't think that's what you mean here, so please reformulate.*

**A:** Thank you very much for your comment. We believe the "Coefficient of Variation" is correct here. As we said in Line 310-311, "the method derives a common percentage change in all flood peaks given changing human factors, which means no changes in coefficients of variation." For example, suppose we have random variable *Z*, a percentage change in *Z* (e.g., a 10% decrease) makes *Z* to be 0.9*Z*. In this case, the coefficient of variation is Std.(0.9*Z*)/Mean(0.9*Z*)= Std.(*Z*)/Mean(*Z*). Therefore, the causal effect in our study does not allow the changes in the coefficient of variation of floods.

**R2C13:** *L294. You provide a good illustrative example of the models inability to capture heterogeneous treatment in time, but here would also be a good opportunity to give an example of a heterogeneous treatment in space (i.e. a scenario where cropland*

*might persistently have a stronger effect on flow peaks in some locations than in other ).*
*That would contribute alleviating my first major concern, above.*

**A:** Thank you very much for your comment. Omitting spatially heterogeneous effects has been mentioned as the first limitation of the method in Line 305 as "no interaction terms between human factors and regional or individual characteristics". To make it clear, we have changed this sentence to "no interaction terms between human factors and regional or individual characteristics that produce significant spatially heterogeneous effects". We have also provided an example of spatially heterogeneous effects in Line 305-307 as "These assumptions may be violated in some cases. For example, the effect of urbanization on floods may be larger in regions with higher soil permeability, which means spatially heterogeneous effects may be nonnegligible."

**R2C14:** *SI. Please add a descriptive statistics table with key stats on all the considered variable across your sample.*

**A:** We have added the table in the revised manuscript. Please also see R2C4.

*Marc Muller*

*References Blum, A. Et al (2020) Causal Effect of Impervious Cover on Annual Flood Magnitude in the United States, GRL*

*Davenport et al. 2020 Flood Size Increases Nonlinearly Across the Western United States in Response to Lower Snow-Precipitation Ratios, WRR*

*Muller, M.F. and Thompson, S.E. (2015) "TopREML: a topological restricted maximum likelihood approach to regionalize trended runoff signatures in stream networks", HESS*

---

## Author Response (AR2)

Dear Editor and Reviewers,

Thank you again for all the suggestions that help us to improve our study. We have revised our manuscript thoroughly according to the comments. Please see the replies to the reviewers' comments below. In the response, the blue texts are the comments and the green texts are the quotes of the manuscript.

On behalf of all co-authors,
Wencong Yang

**Reviewer #1 Comment 1:** (hereafter referred to as R1C1, R1C2…) *The authors have made many improvements to the paper and have addressed most of my concerns. My remaining concern relates to the amount of underlying variability that is used to estimate the panel regression coefficients. The authors should communicate the underlying variability (or lack of) more clearly, so that the reader is able to properly evaluate how the results may or may not apply to other regions or catchments. Below, I have explained my concern in detail and how I believe it could be addressed in a straightforward manner.*

**A:** Thank you for your constructive comments. We have carefully considered your suggestions and revised the paper. In the revision, we compare the characteristics (*Urban*, $\Delta Urban$, *RI*, and $\Delta RI$) of catchment samples used for the regression and the flood change attribution and find no substantial difference. Therefore, we infer that the regression coefficients can be applied on a national scale. In addition, using bootstrapping to derive coefficient intervals has already accounted for the sampling uncertainties to a certain extent. Nonetheless, we admit that the catchment samples used for the regression and the flood change attribution may not be completely homogeneous in terms of all catchment characteristics (e.g., climate, topography). Therefore, we added a statement in the discussion part for readers to transfer the results to other regions with caution. Details of the revision are presented in the following answers.

**R1C2:** *Panel regressions estimate the effect of a within-catchment change by taking advantage of panel datasets, which have variation in both space and time. This requires that there are temporal variations in urban area, RI, and crop area within catchments. As I suspected, and as shown in Table 1, a relatively small portion of catchments experience changes in RI over 1992-2017. For example, it appears that less than half of 207 watersheds (or <103 catchments) had changes in RI. Thus, the*

*estimated effect of RI on Q will be almost entirely based on the effects within this small subset of catchments. My concern is that this subset of catchments may not be a representative subset of the overall 1,625 catchments (in terms of size, geographic distribution, topography, climate, etc).*

**A:** Thank you very much for your critical comment. We agree with you that the regression coefficients rely heavily on the catchment samples with changes in *Urban*, *Crop*, and *RI*. Therefore, in the revision, we compare the frequency distributions of *Urban*, Δ*Urban*, *RI*, and Δ*RI* for catchments used for the regression (Fig. 3) and the flood change attribution (Fig. 6 and Fig. 7). The comparisons are presented in Fig. A2 and Fig. A3 of the appendix and also as follows. We only focus on the catchments with Δ*Urban*>0 and Δ*RI*>0. According to these new figures, the catchments used for the regression are not substantially different from the catchments for the flood change attribution. Therefore, the catchments used for the regression are representative of a national-scale study in terms of the human factors that influence floods. In the revised manuscript, we added the contents about the comparisons above in Line 244-248 as "To avoid sample heterogeneity between these 1625 catchments and the 757 catchments used for regression, we compared the frequency distribution of $Urban$ and $\Delta Urban$ for catchments with $\Delta Urban > 0$ in the two samples of catchments in Fig. A2. Since there is no substantial difference between the two distributions in Fig. A2, the sensitivity of $Q$ to $Urban$, which is derived from the 757 catchments, can be used to infer the changes in $Q$ due to $\Delta Urban$ in those 1625 catchments." and Line 263-267 as "Similar to Fig. 6, to avoid sample heterogeneity between these 536 catchments and the 757 catchments used for regression, we compared the frequency distribution of $RI$ and $\Delta RI$ for catchments with $\Delta RI > 0$ in the two samples of catchments in Fig. A3. Since there is no substantial difference between the two distributions in Fig. A3, the sensitivity of $Q$ to $RI$, which is derived from the 757 catchments, can be used to infer the changes in $Q$ due to $\Delta RI$ in those 536 catchments."

[Figure]

**Figure A2. Frequency distribution of _Urban_ and Δ_Urban_ in catchments with Δ_Urban_ > 0 from two catchment sets: the one used for the regression in Fig. 3(d) (green) and the one used for the flood change attribution in Fig. 6 (red).**

[Figure]

**Figure A3. Frequency distribution of _RI_ and Δ_RI_ in catchments with Δ_RI_ > 0 from two catchment sets: the one used for the regression in Fig. 3(f) (green) and the one used for the flood change attribution in Fig. 7 (red).**

However, the catchments used for the regression may not be representative if we consider more catchment characteristics, such as climate, topography, and so on. It is beyond the scope of this study to determine whether two catchment samples are homogeneous in terms of a comprehensive set of characteristics. Therefore, in the revision, we state the risk of transferring the results of this study to other regions in the discussion as "Caution is required to interpret the flood changes attributed to urbanization and dam constructions on a national scale because the sensitivity of

floods to these factors is derived from a subset of catchments. Although the catchments used for sensitivity calculation and the ones used for flood change attribution have similar frequency distributions of urban areas and reservoir indexes in Fig. A2 and A3, these different sets of catchments may not be completely homogeneous in terms of all characteristics (topography, climate, etc.). Moreover, one should also be cautious to apply the sensitivity results to other regions such as catchments in other countries." (Please see Line 370-375)

**R1C3:** *Similarly, from Table 1, it appears that more than 75% of the 757 catchments had changes in urban area of less than 1%, and more than 50% of catchments had changes in crop area of less than +/– 1%. The few watersheds that had larger changes in urban area or crop area would have a larger influence on the estimated coefficients. Overall, this adds uncertainty for drawing conclusions about effects of RI, crop area, and urban area across the full region, if the regression coefficients are heavily influenced by a few catchments.*
**A:** Thank you for your important comment. Please see the answer to R1C2.

**R1C4:** *I agree with the authors' choice to use bootstrapping to estimate regression uncertainty. I think there are two additional aspects needed to properly convey the underlying variability and the implications for the analysis:*
*1. Clarify the number of catchments with variability in each of the causal factors during the 1992-2017 period, as well as the distribution and characteristics of these catchments compared to the overall region. This could be accomplished with an additional table (or a revised version of Table 1), as well as some clarifications in the text. Fig. 3 is helpful, and it would also be helpful to see the spatial distribution of $\Delta RI$, $\Delta\%Urban$, and $\Delta\%Crop$. If there are large differences between the subset of catchments and the full set of 1,625 (or 2,739) catchments, the authors should add a caveat about applying results from the subset to the full set.*
**A:** Thank you very much for your valuable comment. Following your suggestions, in the revision, we clarify the number of catchments with variability in each of the causal factors in Line 195-197 as "Catchments with changes in $Urban$, $Crop$, and $RI$ have large impacts on estimating regression coefficients. The numbers of catchments with $\Delta Urban > 0$, $\Delta Crop > 0$, and $\Delta RI > 0$ are 656, 351, and 64, respectively.". In addition, we add three subfigures to the original Fig. 3 to show $\Delta Urban$, $\Delta Crop$, and $\Delta RI$.

[Figure]

**Figure 3. Spatial distribution of catchment characteristics in 757 independent catchments. (a) Urban percentages ($Urban$), (b) cropland percentages ($Crop$), and (c) reservoir indexes ($RI$) in their last years with available flood data. The changes of (d) urban areas ($\Delta Urban$), (e) cropland areas ($\Delta Crop$), and (f) reservoir indexes ($\Delta RI$) in 1992-2017.**

As we mentioned in the answer to R1C2, Fig. A2 suggests no substantial difference in the frequency distributions of *Urban* and $\Delta$*Urban* between the 757 independent catchments and the 1625 catchments used for detecting *Urban*-induced flood changes. Nonetheless, we add a caveat in the discussion (Line 370-375) for rigorousness, as we stated in the answer to R1C2.

**R1C5:** *2. In the case of RI (where different time periods are used to estimate the regression vs. cumulative effects), the authors should compare the amount of within-catchment variability during the period used for fitting the regression (1992-2017) to the RI changes during the period used to estimate cumulative causal effects (1960-2017). This could be accomplished in a table, or through clarification in the text. It is possible that changes in RI over this longer period are much larger than over the shorter period, and thus there may be higher uncertainty when extrapolating to these larger changes. If this is the case, I think it deserves a statement or caveat in the text.*

**A:** Thank you very much for your valuable comment. As we mentioned in the answer to R1C2, Fig. A3 suggests no substantial difference in the frequency distributions of

*RI* and Δ*RI* between the 757 independent catchments in 1992-2017 and the 536 catchments in 1960-2017. Nonetheless, we add a caveat in the discussion (Line 370-375) for rigorousness, as we stated in the answer to R1C2.

*Additional Comments:*

**R1C6:** *L164: "1. There are no other important time-varying sub-regional variables that significantly affect both human factors and floods". I think the assumption would more accurately be stated as: There are no other time-varying sub-regional variables that are correlated with human factors and affect floods.*

**A:** Thank you very much for your comment. We have changed the sentence to "There are no other time-varying sub-regional variables that correlate with both human factors and floods". (Please see Line 141-142)

**R1C7:** *Figure 8: I think this analysis should be restricted to the common period when both dam and land type data are available (1991-2017). Otherwise, potential effects of Δ%Urban in the early part of the period are unknown. Additionally, I am confused by the authors' choice to only examine basins where RI=0 and %Urban=0 at the beginning of the time period. Any catchments that have no change in RI and %Urban would be free of effects from these factors over the time period used to calculate trends in Q (regardless of initial RI or %Urban values). I don't think that the current analysis is wrong, per se, but a more comprehensive version of Fig. 8 would include all catchments where ΔRI and Δ%Urban yield expected changes in Q of less than 10% (regardless of initial values), and is something the authors should consider.*

**A:** Thank you very much. The purpose of Fig. 8 is to examine the unexplained flood changes in a long period (≥30 years). Long-term changes in floods dated back to the 1960s provide valuable information for studies related to the impacts of climate change or other factors on floods. Since we do not have land cover data before 1992, the strategy to select catchments is to filter all catchments that are potential to have large Δ*Urban* values in 1960-2017. In other words, the catchments with limited values of *Urban* are a subset of catchments with limited values of Δ*Urban*. Therefore, the selection guarantees the catchments are free from the impact of *Urban*, but does not guarantee all catchments that are free from the impact of *Urban* in 1960-2017 will be included. You indeed provide a good idea to redraw Fig. 8, but we aim to show the flood changes in a longer period and therefore we prefer to keep this figure.